# Efficient and Accurate Gradients for Neural SDEs

**Patrick Kidger**[1]  **James Foster**[1]  **Xuechen Li**[2]  **Terry Lyons**[1]

[1] University of Oxford; The Alan Turing Insitute  [2] Stanford
{kidger, foster, tlyons}@maths.ox.ac.uk
lxuechen@cs.toronto.edu

## Abstract

Neural SDEs combine many of the best qualities of both RNNs and SDEs: memory efficient training, high-capacity function approximation, and strong priors on model space. This makes them a natural choice for modelling many types of temporal dynamics. Training a Neural SDE (either as a VAE or as a GAN) requires backprop-agating through an SDE solve. This may be done by solving a backwards-in-time SDE whose solution is the desired parameter gradients. However, this has previously suffered from severe speed and accuracy issues, due to high computational cost and numerical truncation errors. Here, we overcome these issues through several technical innovations. First, we introduce the *reversible Heun method*. This is a new SDE solver that is *algebraically reversible*: eliminating numerical gradient errors, and the first such solver of which we are aware. Moreover it requires half as many function evaluations as comparable solvers, giving up to a $1.98\times$ speedup. Second, we introduce the *Brownian Interval*: a new, fast, memory efficient, and exact way of sampling *and reconstructing* Brownian motion. With this we obtain up to a $10.6\times$ speed improvement over previous techniques, which in contrast are both approximate and relatively slow. Third, when specifically training Neural SDEs as GANs (Kidger et al. 2021), we demonstrate how SDE-GANs may be trained through careful weight clipping and choice of activation function. This reduces computational cost (giving up to a $1.87\times$ speedup) and removes the numer-ical truncation errors associated with gradient penalty. Altogether, we outperform the state-of-the-art by substantial margins, with respect to training speed, and with respect to classification, prediction, and MMD test metrics. We have contributed implementations of all of our techniques to the `torchsde` library to help facilitate their adoption.

## 1  Introduction

**Stochastic differential equations** Stochastic differential equations have seen widespread use in the mathematical modelling of random phenomena, such as particle systems [1], financial markets [2], population dynamics [3], and genetics [4]. Featuring inherent randomness, then in modern machine learning parlance SDEs are generative models.

Such models have typically been constructed theoretically, and are usually relatively simple. For example the Black–Scholes equation, widely used to model asset prices in financial markets, has only two scalar parameters: a fixed drift and a fixed diffusion [5].

**Neural stochastic differential equations** Neural stochastic differential equations offer a shift in this paradigm. By parameterising the drift and diffusion of an SDE as neural networks, then modelling capacity is greatly increased, and theoretically arbitrary SDEs may be approximated. (By the universal approximation theorem for neural networks [6, 7].) Several authors have now studied or introduced Neural SDEs; [8, 9, 10, 11, 12, 13, 14, 15, 16] amongst others.

35th Conference on Neural Information Processing Systems (NeurIPS 2021).

**Connections to recurrent neural networks**   A numerically discretised (Neural) SDE may be interpreted as an RNN (featuring a residual connection), whose input is random noise – Brownian motion – and whose output is a generated sample. Subject to a suitable loss function between distributions, such as the KL divergence [15] or Wasserstein distance [16], this may then simply be backpropagated through in the usual way.

**Generative time series models**   SDEs are naturally random. In modern machine learning parlance they are thus generative models. As such we treat Neural SDEs as *generative time series models*.

The (recurrent) neural network-like structure offers high-capacity function approximation, whilst the SDE-like structure offers strong priors on model space, memory efficiency, and deep theoretical connections to a well-understood literature. Relative to the classical SDE literature, Neural SDEs have essentially unprecedented modelling capacity.

(Generative) time series models are of classical interest, with forecasting models such as Holt–Winters [17, 18], ARMA [19] and so on. It has also attracted much recent interest with (besides Neural SDEs) the development of models such as Time Series GAN [20], Latent ODEs [21], GRU-ODE-Bayes [22], ODE$^2$VAE [23], CTFPs [24], Neural ODE Processes [25] and Neural Jump ODEs [26].

## 1.1   Contributions

We study backpropagation through SDE solvers, in particular to train Neural SDEs, via continuous adjoint methods. We introduce several technical innovations to improve both model performance and the speed of training: in particular to reduce numerical gradient errors to almost zero.

First, we introduce the *reversible Heun method*: a new SDE solver, constructed to be *algebraically reversible*. By matching the truncation errors of the forward and backward passes, the gradients computed via continuous adjoint method are precisely those of the numerical discretisation of the forward pass. This overcomes the typical greatest limitation of continuous adjoint methods – and to the best of our knowledge, is the first algebraically reversible SDE solver to have been developed.

After that, we introduce the *Brownian Interval* as a new way of sampling *and reconstructing* Brownian motion. It is fast, memory efficient and exact. It has an average (modal) time complexity of $\mathcal{O}\left(1\right)$, and consumes only $\mathcal{O}\left(1\right)$ GPU memory. This is contrast to previous techniques requiring either $\mathcal{O}\left(T\right)$ memory, or a choice of approximation error $\varepsilon \ll 1$ and then a time complexity of $\mathcal{O}\left(\log(1/\varepsilon)\right)$.

Finally, we demonstrate how the Lipschitz condition for the discriminator of an SDE-GAN may be imposed without gradient penalties – instead using careful clipping and the LipSwish activation function – so as to overcome their previous incompatibility with continuous adjoint methods.

Overall, multiple technical innovations provide substantial improvements over the state-of-the-art with respect to training speed, and with respect to classification, prediction, and MMD test metrics.

## 2   Background

### 2.1   Neural SDE construction

**Certain minimal amount of structure**   Following Kidger et al. [16], we construct Neural SDEs with a certain minimal amount of structure. Let $T > 0$ be fixed and suppose we wish to model a path-valued random variable $Y_{\text{true}} \colon [0, T] \to \mathbb{R}^y$. The size of $y$ is the dimensionality of the data.[1]

Let $W \colon [0, T] \to \mathbb{R}^w$ be a $w$-dimensional Brownian motion, and let $V \sim \mathcal{N}(0, I_v,)$ be drawn from a $v$-dimensional standard multivariate normal. The values $w, v$ are hyperparameters describing the size of the noise. Let

$$\zeta_\theta \colon \mathbb{R}^v \to \mathbb{R}^x, \quad \mu_\theta \colon [0, T] \times \mathbb{R}^x \to \mathbb{R}^x, \quad \sigma_\theta \colon [0, T] \times \mathbb{R}^x \to \mathbb{R}^{x \times w}, \quad \ell_\theta \colon \mathbb{R}^x \to \mathbb{R}^y,$$

where $\zeta_\theta$, $\mu_\theta$ and $\sigma_\theta$ are neural networks and $\ell_\theta$ is affine. Collectively these are parameterised by $\theta$. The dimension $x$ is a hyperparameter describing the size of the hidden state.

---

[1]In practice we will typically observe some discretised time series sampled from $Y_{\text{true}}$. For ease of presentation we will neglect this detail for now and will return to it in Section 2.3.

We consider Neural SDEs as models of the form

$$X_0 = \zeta_\theta(V), \qquad \mathrm{d}X_t = \mu_\theta(t, X_t)\,\mathrm{d}t + \sigma_\theta(t, X_t) \circ \mathrm{d}W_t, \qquad Y_t = \ell_\theta(X_t), \qquad (1)$$

for $t \in [0, T]$, with $X : [0, T] \to \mathbb{R}^x$ the (strong) solution to the SDE.[2] The solution $X$ is guaranteed to exist given mild conditions: that $\mu_\theta$, $\sigma_\theta$ are Lipschitz, and that $\mathbb{E}_V\left[\zeta_\theta(V)^2\right] < \infty$.

We seek to train this model such that $Y \overset{\mathrm{d}}{\approx} Y_{\text{true}}$. That is to say, the model $Y$ should have approximately the same distribution as the target $Y_{\text{true}}$, for some notion of approximate. (For example, to be similar with respect to the Wasserstein distance).

**RNNs as discretised SDEs** The minimal amount of structure is chosen to parallel RNNs. The solution $X$ may be interpreted as hidden state, and the $\ell_\theta$ maps the hidden state to the output of the model. In Appendix A we provide sample PyTorch [27] code computing a discretised SDE according to the Euler–Maruyama method. The result is an RNN consuming random noise as input.

## 2.2 Training criteria for Neural SDEs

Equation (1) produces a random variable $Y : [0, T] \to \mathbb{R}^y$ implicitly depending on parameters $\theta$. This model must still be fit to data. This may be done by optimising a distance between the probability distributions (laws) for $Y$ and $Y_{\text{true}}$.

**SDE-GANs** The Wasserstein distance may be used by constructing a discriminator and training adversarially, as in Kidger et al. [16]. Let $F_\phi(Y) = m_\phi \cdot H_T$, where

$$H_0 = \xi_\phi(Y_0), \qquad \mathrm{d}H_t = f_\phi(t, H_t)\,\mathrm{d}t + g_\phi(t, H_t) \circ \mathrm{d}Y_t, \qquad (2)$$

for suitable neural networks $\xi_\phi, f_\phi, g_\phi$ and vector $m_\phi$. This is a deterministic function of the generated sample $Y$. Here $\cdot$ denotes a dot product. They then train with respect to

$$\min_\theta \max_\phi \left( \mathbb{E}_Y\left[F_\phi(Y)\right] - \mathbb{E}_{Y_{\text{true}}}\left[F_\phi(Y_{\text{true}})\right] \right). \qquad (3)$$

See Appendix B for additional details on this approach, and in particular how it generalises the classical approach to fitting (calibrating) SDEs.

**Latent SDEs** Li et al. [15] instead optimise a KL divergence. This consists of constructing an auxiliary process $\widehat{X}$ with drift $\nu_\phi$ parameterised by $\phi$, and optimising an expression of the form

$$\min_{\theta, \phi} \mathbb{E}_{W, Y_{\text{true}}}\left[ \int_0^T (Y_{\text{true}, t} - \ell_\theta(\widehat{X}_t))^2 + \frac{1}{2} \left\| (\sigma_\theta(t, \widehat{X}_t))^{-1}(\mu_\theta(t, \widehat{X}_t) - \nu_\phi(t, \widehat{X}_t, Y_{\text{true}})) \right\|_2^2 \mathrm{d}t \right]. \qquad (4)$$

The full construction is moderately technical; see Appendix B for further details.

## 2.3 Discretised observations

Observations of $Y_{\text{true}}$ are typically a discrete time series, rather than a true continuous-time path. This is not a serious hurdle. If training an SDE-GAN, then equation (2) may be evaluated on an interpolation $Y_{\text{true}}$ of the observed data. If training a Latent SDE, then $\nu_\phi$ in equation (4) may depend explicitly on the discretised $Y_{\text{true}}$.

## 2.4 Backpropagation through SDE solves

Whether the loss for our generated sample $Y$ is produced via a Latent SDE or via the discriminator of an SDE-GAN, it is still required to backpropagate from the loss to the parameters $\theta, \phi$.

Here we use *the continuous adjoint method*. Also known as simply 'the adjoint method', or 'optimise-then-discretise', this has recently attracted much attention in the modern literature on neural differential equations. This exploits the reversibility of a differential equation: as with invertible neural

---

[2]The notation '$\circ\,\mathrm{d}W_t$' denotes Stratonovich integration. Itô is less efficient; see Appendix C.

networks [28], intermediate computations such as $X_t$ for $t < T$ are reconstructed from output computations, so that they do not need to be held in memory.

Given some Stratonovich SDE

$$\mathrm{d}Z_t = \mu(t, Z_t)\,\mathrm{d}t + \sigma(t, Z_t) \circ \mathrm{d}W_t \quad \text{for } t \in [0, T], \tag{5}$$

and a loss $L\colon \mathbb{R}^z \to \mathbb{R}$ on its terminal value $Z_T$, then the adjoint process $A_t = \mathrm{d}L(Z_T)/\mathrm{d}Z_t \in \mathbb{R}^z$ is a (strong) solution to

$$\mathrm{d}A_t^i = -A_t^j \frac{\partial \mu^j}{\partial Z^i}(t, Z_t)\,\mathrm{d}t - A_t^j \frac{\partial \sigma^{j,k}}{\partial Z^i}(t, Z_t) \circ \mathrm{d}W_t^k, \tag{6}$$

which in particular uses the same Brownian motion $W$ as on the forward pass. Equations (5) and (6) may be combined into a single SDE and solved backwards-in-time[3] from $t = T$ to $t = 0$, starting from $Z_T = Z_T$ (computed on the forward pass of equation (5)) and $A_T = L(Z_T)/\mathrm{d}Z_T$. Then $A_0 = \mathrm{d}L(Z_T)/Z_0$ is the desired backpropagated gradient.

Note that we assumed here that the loss $L$ acts only on $Z_T$, not all of $Z$. This is not an issue in practice. In both equations (3) and (4), the loss is an integral. As such it may be computed as part of $Z$ in a single SDE solve. This outputs a value at time $T$, the operation $L$ may simply extract this value from $Z_T$, and then backpropagation may proceeed as described here.

The main issue is that the two numerical approximations to $Z_t$, computed in the forward and backward passes of equation (5), are different. This means that the $Z_t$ used as an input in equation (6) has some discrepancy from the forward calculation, and the gradients $A_0$ suffer some error as a result. (Often exacerbating an already tricky training procedure, such as the adversarial training of SDE-GANs.)

See Appendix C for further discussion on how an SDE solve may be backpropagated through.

## 2.5 Alternate constructions

There are other uses for Neural SDEs, beyond our scope here. For example Song et al. [30] combine SDEs with score-matching, and Xu et al. [31] use SDEs to represent Bayesian uncertainty over parameters. The techniques introduced in this paper will apply to any backpropagation through an SDE solve.

## 3 Reversible Heun method

We introduce a new SDE solver, which we refer to as the *reversible Heun method*. Its key property is algebraic reversibility; moreover to the best of our knowledge it is the first SDE solver to exhibit this property.

To fix notation, we consider solving the Stratonovich SDE

$$\mathrm{d}Z_t = \mu(t, Z_t)\,\mathrm{d}t + \sigma(t, Z_t) \circ \mathrm{d}W_t, \tag{7}$$

with known initial condition $Z_0$.

**Solver** We begin by selecting a step size $\Delta t$, and initialising $t_0 = 0$, $z_0 = \widehat{z}_0 = Z_0$, $\mu_0 = \mu(0, Z_0)$ and $\sigma_0 = \sigma(0, Z_0)$. Let $W$ denote a single sample path of Brownian motion. It is important that the same sample be used for both the forward and backward passes of the algorithm; computationally this may be accomplished by taking $W$ to be a Brownian Interval, which we will introduce in Section 4.

---

**Algorithm 1:** Forward pass

**Input:** $t_n, z_n, \widehat{z}_n, \mu_n, \sigma_n, \Delta t, W$

$t_{n+1} = t_n + \Delta t$

$\Delta W_n = W_{t_{n+1}} - W_{t_n}$

$\widehat{z}_{n+1} = 2z_n - \widehat{z}_n + \mu_n \Delta t + \sigma_n \Delta W_n$

$\mu_{n+1} = \mu(t_{n+1}, \widehat{z}_{n+1})$

$\sigma_{n+1} = \sigma(t_{n+1}, \widehat{z}_{n+1})$

$z_{n+1} = z_n + \dfrac{1}{2}(\mu_n + \mu_{n+1})\Delta t$
$\qquad\quad + \dfrac{1}{2}(\sigma_n + \sigma_{n+1})\Delta W_n$

**Output:** $t_{n+1}, z_{n+1}, \widehat{z}_{n+1}, \mu_{n+1}, \sigma_{n+1}$

---

We then iterate Algorithm 1. Suppose $T = N\Delta t$ so that $z_N, \widehat{z}_N, \mu_N, \sigma_N$ are the final output. Then $z_N \approx Z_T$ is returned, whilst $z_N, \widehat{z}_N, \mu_N, \sigma_N$ are all retained for the backward pass.

Nothing else need be saved in memory for the backward pass: in particular no intermediate computations, as would otherwise be typical.

---

[3]Li et al. [15] give rigorous meaning to this via two-sided filtrations; for the reader familiar with rough path theory then Kidger et al. [29, Appendix A] also give a pathwise interpretation. The reader familiar with neither of these should feel free to intuitively treat Stratonovich (but not Itô) SDEs like ODEs.

**Algebraic reversibility**  The key advantage of the reversible Heun method, and the motivating reason for its use alongside continuous-time adjoint methods, is that it is algebraically reversible. That is, it is possible to reconstruct $(z_n, \widehat{z}_n, \mu_n, \sigma_n)$ from $(z_{n+1}, \widehat{z}_{n+1}, \mu_{n+1}, \sigma_{n+1})$ in closed form. (And without a fixed-point iteration.)

This crucial property will mean that it is possible to backpropagate through the SDE solve, such that the gradients obtained via the continuous adjoint method (equation (6)) *exactly match* the (discretise-then-optimise) gradients obtained by autodifferentiating the numerically discretised forward pass.

In doing so, one of the greatest limitations of continuous adjoint methods is overcome.

To the best of our knowledge, the reversible Heun method is the first algebraically reversible SDE solver.

**Computational efficiency**  A further advantage of the reversible Heun method is computational efficiency. The method requires only a single function evaluation (of both the drift and diffusion) per step. This is in contrast to other Stratonovich solvers (such as the midpoint method or regular Heun's method), which require two function evaluations per step.

---

**Algorithm 2:** Backward pass

**Input:** $t_{n+1}, z_{n+1}, \widehat{z}_{n+1}, \mu_{n+1}, \sigma_{n+1}, \Delta t, W,$
$$\frac{\partial L(Z_T)}{\partial z_{n+1}}, \frac{\partial L(Z_T)}{\partial \widehat{z}_{n+1}}, \frac{\partial L(Z_T)}{\partial \mu_{n+1}}, \frac{\partial L(Z_T)}{\partial \sigma_{n+1}}$$

\# Reverse step
$$t_n = t_{n+1} - \Delta t$$
$$\Delta W_n = W_{t_{n+1}} - W_{t_n}$$
$$\widehat{z}_n = 2z_{n+1} - \widehat{z}_{n+1} - \mu_{n+1}\Delta t - \sigma_{n+1}\Delta W_n$$
$$\mu_n = \mu(t_n, \widehat{z}_n)$$
$$\sigma_n = \sigma(t_n, \widehat{z}_n)$$
$$z_n = z_{n+1} - \frac{1}{2}(\mu_n + \mu_{n+1})\Delta t$$
$$\qquad - \frac{1}{2}(\sigma_n + \sigma_{n+1})\Delta W_n$$

\# Local forward
$$z_{n+1}, \widehat{z}_{n+1}, \mu_{n+1}, \sigma_{n+1} = \text{Forward}(t_n, z_n, \widehat{z}_n, \mu_n,$$
$$\sigma_n, \Delta t, W)$$

\# Local backward
$$\frac{\partial L(Z_T)}{\partial(z_n, \widehat{z}_n, \mu_n, \sigma_n)} = \frac{\partial L(Z_T)}{\partial(z_{n+1}, \widehat{z}_{n+1}, \mu_{n+1}, \sigma_{n+1})}$$
$$\cdot \frac{\partial(z_{n+1}, \widehat{z}_{n+1}, \mu_{n+1}, \sigma_{n+1})}{\partial(z_n, \widehat{z}_n, \mu_n, \sigma_n)}$$

**Output:** $t_n, z_n, \widehat{z}_n, \mu_n, \sigma_n,$
$$\frac{\partial L(Z_T)}{\partial z_n}, \frac{\partial L(Z_T)}{\partial \widehat{z}_n}, \frac{\partial L(Z_T)}{\partial \mu_n}, \frac{\partial L(Z_T)}{\partial \sigma_n}$$

---

**Convergence of the solver**  When applied to the Stratonovich SDE (7), the reversible Heun method exhibits strong convergence of order 0.5; the same as the usual Heun's method.

**Theorem.** *Let $\{z_n\}$ denote the numerical solution of (7) obtained by Algorithm 1 with a constant step size $\Delta t$ and assume sufficient regularity of $\mu$ and $\sigma$. Then there exists a constant $C > 0$ so that*

$$\mathbb{E}\Big[\big\|z_N - Z_T\big\|_2\Big] \leq C\sqrt{\Delta t}\,,$$

*for small $\Delta t$. That is, strong convergence of order 0.5. If $\sigma$ is constant, then this improves to order 1.*

The key idea in the proof is to consider two adjacent steps of the SDE solver. Then the update $\widehat{z}_n \mapsto \widehat{z}_{n+2}$ becomes a step of a midpoint method, whilst $z_n \mapsto z_{n+1}$ is similar to Heun's method. This makes it possible to show that $\{\widehat{z}_n\}$ and $\{z_n\}$ stay close together: $\mathbb{E}\big[\|z_n - \widehat{z}_n\|_2^4\big] \sim O(\Delta t^2)$. With this $\mathbb{L}_4$ bound on $z - \widehat{z}$, we can then apply standard lines of argument from the numerical SDE literature. Chaining together local mean squared error estimates, we obtain $\mathbb{E}\big[\|z_N - Z_T\|_2^2\big] \sim O(\Delta t)$.

See Appendix D for the full proof. We additionally consider stability in the ODE setting. Whilst the method is not A-stable, we do show it has the same absolute stability region for a linear test equation as the (reversible) asynchronous leapfrog integrator proposed for Neural ODEs in Zhuang et al. [32].

**Precise gradients**  The backward pass is shown in Algorithm 2. As the same numerical solution $\{z_n\}$ is recovered on both the forward and backward passes – exhibiting the same truncation errors – then the computed gradients are precisely the (discretise-then-optimise) gradients of the numerical discretisation of the forward pass.

Each $\partial L(Z_T)/\partial z_n \approx A_{n\Delta t}$, where $A$ is the adjoint variable of equation (6).

This is unlike the case of solving equation (6) via standard numerical techniques, for which small or adaptive step sizes are necessary to obtain useful gradients [15].

Table 1: SDE-GAN on weights dataset; Latent SDE on air quality dataset. Mean $\pm$ standard deviation averaged over three runs.

| Dataset, Solver | Label classification accuracy (%) | MMD ($\times 10^{-2}$) | Training time |
|---|---|---|---|
| Weights, Midpoint | — | $4.38 \pm 0.67$ | $5.12 \pm 0.01$ days |
| Weights, Reversible Heun | — | $\mathbf{1.75 \pm 0.3}$ | $\mathbf{2.59 \pm 0.05}$ days |
| Air quality, Midpoint | $46.3 \pm 5.1$ | $0.591 \pm 0.206$ | $5.58 \pm 0.54$ hours |
| Air quality, Reversible Heun | $\mathbf{49.2 \pm 0.02}$ | $\mathbf{0.472 \pm 0.290}$ | $\mathbf{4.47 \pm 0.31}$ hours |

## 3.1 Experiments

We validate the empirical performance of the reversible Heun method. For space, we present abbreviated details and results here. See Appendix F for details of the hyperparameter optimisation procedure, test metric definitions, and so on, and for further results on additional datasets and additional metrics.

**Versus midpoint**  We begin by comparing the reversible Heun method with the midpoint method, which also converges to the Stratonovich solution. We train an SDE-GAN on a dataset of weight trajectories evolving under stochastic gradient descent, and train a Latent SDE on a dataset of air quality over Beijing.

See Table 1. Due to the reduced number of vector field evaluations, we find that training speed roughly doubles ($1.98\times$) on the weights dataset, whilst its numerically precise gradients substantially improve the test metrics (comparing generated samples to a held-out test set). Similar behaviour is observed on the air quality dataset, with substantial test metric improvements and a training speed improvement of $1.25\times$.

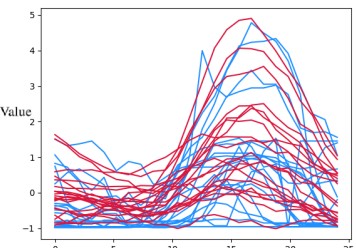

Figure 1: Samples (red) from Latent SDE on $O_3$ ozone channel of air quality dataset (blue).

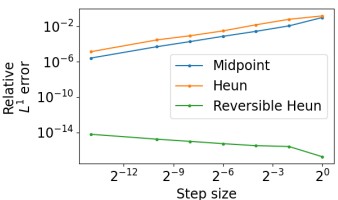

Figure 2: Relative error in gradient calculation.

**Samples**  We verify that samples from a model using reversible Heun resemble that of the original dataset: in Figure 1 we show the Latent SDE on the ozone concentration over Beijing.

**Gradient error**  We investigate the numerical error made in solving (6), compared to the (discretise-then-optimise) gradients of the numerically discretised forward pass. We fix a test problem (differentiating a small Neural SDE) and vary the step size and solver; see Figure 2. The error made in standard solvers is very large (but does at least decrease with step size). The reversible Heun method produces results accurate to floating point error, unattainable by any standard solver.

## 4  Brownian Interval

Numerically solving an SDE, via the reversible Heun method or via any other numerical solver, requires sampling Brownian motion: this is the input $W$ in Algorithms 1 and 2.

**Brownian bridges**  Mathematically, sampling Brownian motion is straightforward. A fixed-step numerical solver may simply sample independent increments during its time stepping. An adaptive solver (which may reject steps) may use Lévy's Brownian bridge [33] formula to generate increments with the appropriate correlations: letting $W_{a,b} = W_b - W_a \in \mathbb{R}^w$, then for $u < s < t$,

$$W_{u,s} | W_{u,t} = \mathcal{N}\Big( \frac{s-u}{t-u} W_{u,t}, \frac{(t-s)(s-u)}{t-u} I_w \Big). \tag{8}$$

**Brownian reconstruction**  However, there are computational difficulties. On the backward pass, the same Brownian sample as the forward pass must be used, and potentially at locations other than were measured on the forward pass [15].

**Time and memory efficiency** The simple but memory intensive approach would be to store every sample made on the forward pass, and then on the backward pass reuse these samples, or sample Brownian noise according to equation (8), as appropriate.

Li et al. [15] instead offer a memory-efficient but time-intensive approach, by introducing the 'Virtual Brownian Tree'. This approximates the real line by a tree of dyadic points. Samples are approximate, and demand deep (slow) traversals of the tree.

**Binary tree of (interval, seed) pairs** In response to this, we introduce the 'Brownian Interval', which offers memory efficiency, exact samples, and fast query times, all at once. The Brownian Interval is built around a binary tree, each node of which is an interval and a random seed.

The tree starts as a stump consisting of the global interval $[0, T]$ and a randomly generated random seed. New leaf nodes are created as observations of the sample are made. For example, making a first query at $[s, t] \subseteq [0, T]$ (an operation that returns $W_{s,t}$) produces the binary tree shown in Figure 3a. Algorithm 4 in Appendix E gives the formal definition of this procedure. Making a subsequent query at $[u, v]$ with $u < s < v < t$ produces Figure 3b. Using a splittable PRNG [34, 35], each child node has a random seed deterministically produced from the seed of its parent.

The tree is thus designed to completely encode the conditional statistics of a sample of Brownian motion: $W_{s,t}, W_{t,u}$ are completely specified by $t, [s, u], W_{s,u}$, equation (8), and the random seed for $[s, u]$.

In principle this now gives a way to compute $W_{s,t}$; calculating $W_{s,u}$ recursively. Naïvely this would be very slow – recursing to the root on every query – which we cover by augmenting the binary tree structure with a fixed-size Least Recently Used (LRU) cache on the computed increments $W_{s,t}$.

---

**Algorithm 3:** Sampling the Brownian Interval

**Type:** Let *Node* denote a 5-tuple consisting of an interval, a seed, and three optional *Node*s, corresponding to the parent node, and two child nodes, respectively. (Optional as the root has no parent and leaves have no children.)

**State:** Binary tree with elements of type *Node*, with root $\widehat{I} = ([0, T], \widehat{s}, *, \widehat{I}_{\texttt{left}}, \widehat{I}_{\texttt{right}})$. A *Node* $\widehat{J}$.

**Input:** Interval $[s, t] \subseteq [0, T]$

```
# The returned 'nodes' is a list of Nodes whose
# intervals partition [s, t]. Practically speaking
# this will usually have only one or two elements.
# Ĵ is a hint for where in the tree we are.
nodes = traverse(Ĵ, [s, t])
```

**def** sample($I$ : *Node*)**:**
  **if** $I$ is $\widehat{I}$ **then**
    |   return $\mathcal{N}(0, T)$ sampled with seed $\widehat{s}$.
  Let $I = ([a, b], s, I_{\texttt{parent}}, I_{\texttt{left}}, I_{\texttt{right}})$
  Let $I_{\texttt{parent}} = ([a_p, b_p], s_p, I_{pp}, I_{lp}, I_{rp})$
  $W_{\texttt{parent}} = $ sample$(I_{\texttt{parent}})$
  **if** $I_i$ is $I_{rp}$ **then**
    |   $W_{\texttt{left}} = $ bridge$(a_p, b_p, a, W_{\texttt{parent}}, s)$
    |   return $W_{\texttt{parent}} - W_{\texttt{left}}$
  **else**
    |   return bridge$(a_p, b_p, b, W_{\texttt{parent}}, s)$

sample = LRUCache(sample)

$\widehat{J} \leftarrow$ nodes$[-1]$
$W_{s,t} = \sum_{I \in \text{nodes}}$ sample$(I)$
**Output:** $W_{s,t}$

---

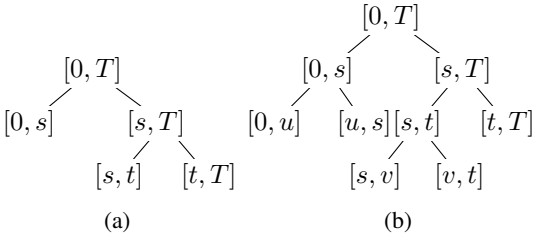

(a)                  (b)

Figure 3: Binary tree of intervals.

See Algorithm 3, where bridge denotes equation (8). The operation traverse traverses the binary tree to find or create the list of nodes whose disjoint union is the interval of interest, and is defined explicitly as Algorithm 4 in Appendix E.

Additionally see Appendix E for various technical considerations and extensions to this algorithm.

**Advantages of the Brownian Interval** The LRU cache ensures that queries have an average-case (modal) time complexity of only $\mathcal{O}(1)$: in SDE solvers, subsequent queries are typically close to (and thus conditional on) previous queries. Even given cache misses all the way up the tree, the worst-case time complexity will only be $\mathcal{O}(\log(1/s))$ in the average step size $s$ of the SDE solver. This is in contrast to the Virtual Brownian Tree, which has an (average or worst-case) time complexity of $\mathcal{O}(\log(1/\varepsilon))$ in the approximation error $\varepsilon \ll s$.

Table 2: V. Brownian Tree against Brownian Interval on speed benchmarks, over 100 subintervals.

| | SDE solve, speed (seconds) | | | Doubly sequential access, speed (seconds) | | |
|---|---|---|---|---|---|---|
| | Size=1 | Size=2560 | Size=32768 | Size=1 | Size=2560 | Size=32768 |
| V. B. Tree | $1.6\times10^0$ | $2.0\times10^0$ | $5.00\times10^2$ | $2.4\times10^{-1}$ | $3.9\times10^{-1}$ | $2.9\times10^0$ |
| B. Interval | $\mathbf{8.2\times10^{-1}}$ | $\mathbf{1.3\times10^0}$ | $\mathbf{4.7\times10^1}$ | $\mathbf{5.0\times10^{-2}}$ | $\mathbf{8.0\times10^{-2}}$ | $\mathbf{3.5\times10^{-1}}$ |

Meanwhile the (GPU) memory cost is only $\mathcal{O}(1)$, corresponding to the fixed and constant size of the LRU cache. There is the small additional cost of storing the tree structure itself, but this is held in CPU memory, which for practical purposes is essentially infinite. This is in contrast to simply holding all the Brownian samples in memory, which has a memory cost of $\mathcal{O}(T)$.

Finally, queries are exact because the tree aligns with the query points. This is contrast to the Virtual Brownian Tree, which only produces samples up to some discretisation of the real line at resolution $\varepsilon$.

### 4.1 Experiments

We benchmark the performance of the Brownian Interval against the Virtual Brownian Tree considered in Li et al. [15]. We include benchmarks corresponding to varying batch sizes, number of sample intervals, and access patterns. For space, just a subset of results are shown. Precise experimental details and further results are available in Appendix F.

See Table 2. We see that the Brownian Interval is uniformly faster than the Virtual Brownian Tree, ranging from $1.5\times$ faster on smaller problems to $10.6\times$ faster on larger problems. Moreover these speed gains are despite the Brownian Interval being written in Python, whilst the Virtual Brownian Tree is carefully optimised and written in C++.

## 5 Training SDE-GANs without gradient penalty

Kidger et al. [16] train SDEs as GANs, as discussed in Section 2.2, using a neural CDE as a discriminator as in equation (2). They found that only gradient penalty [36] was suitable to enforce the Lipschitz condition, given the recurrent structure of the discriminator.

However gradient penalty requires calculating second derivatives (a 'double-backward'). This complicates the use of continuous adjoint methods: the double-continuous-adjoint introduces substantial truncation error; sufficient to obstruct training and requiring small step sizes to resolve.

Here we overcome this limitation, and moreover do so independently of the possibility of obtaining exact double-gradients via the reversible Heun method. For simplicity we now assume throughout that our discriminator vector fields $f_\phi$, $g_\phi$ are MLPs, which is also the choice we make in practice.

**Lipschitz constant one** The key point is that the vector fields $f_\phi$, $g_\phi$ of the discriminator must not only be Lipschitz, but must have Lipschitz constant at most one.

Given vector fields with Lipschitz constant $\lambda$, then the recurrent structure of the discriminator means that the Lipschitz constant of the overall discriminator will be $\mathcal{O}(\lambda^T)$. Ensuring $\lambda \approx 1$ with $\lambda \leq 1$ thus enforces that the overall discriminator is Lipschitz with a reasonable Lipschitz constant.

**Hard constraint** The exponential size of $\mathcal{O}(\lambda^T)$ means that $\lambda$ only slightly greater than one is still insufficient for stable training. We found that this ruled out enforcing $\lambda \leq 1$ via soft constraints, via either spectral normalisation [37] or gradient penalty across just vector field evaluations.

**Clipping** The first part of enforcing this Lipschitz constraint is careful clipping. Considering each linear operation from $\mathbb{R}^a \to \mathbb{R}^b$ as a matrix in $A \in \mathbb{R}^{a \times b}$, then after each gradient update we clip its entries to the region $[-1/b, 1/b]$. Given $x \in \mathbb{R}^a$, then this enforces $\|Ax\|_\infty \leq \|x\|_\infty$.

**LipSwish activation functions** Next we must pick an activation function with Lipschitz constant at most one. It should additionally be at least twice continuously differentiable to ensure convergence of the numerical SDE solver (Appendix D). In particular this rules out the ReLU.

Table 3: SDE-GAN on OU dataset. Mean $\pm$ standard deviation averaged over three runs.

| Solver | Test Metrics | | | |
| --- | --- | --- | --- | --- |
| | Real/fake classification accuracy (%) | Prediction loss | MMD $(\times 10^{-1})$ | Training time (hours) |
| Midpoint w/ gradient penalty | $98.2 \pm 2.4$ | $2.71 \pm 1.03$ | $2.58 \pm 1.81$ | $55.0 \pm 27.7$ |
| Midpoint w/ clipping | $93.9 \pm 6.9$ | $1.65 \pm 0.17$ | $1.03 \pm 0.10$ | $32.5 \pm 12.1$ |
| Reversible Heun w/ clipping | $\mathbf{67.7 \pm 1.1}$ | $\mathbf{1.38 \pm 0.06}$ | $\mathbf{0.45 \pm 0.22}$ | $\mathbf{29.4 \pm 8.9}$ |

There remain several admissible choices; we use the LipSwish activation function introduced by Chen et al. [38], defined as $\rho(x) = 0.909\,x\,\mathrm{sigmoid}(x)$. This was carefully constructed to have Lipschitz constant one, and to be smooth. Moreover the SiLU activation function [39, 40, 41] from which it is derived has been reported as an empirically strong choice.

The overall vector fields $f_\phi$, $g_\phi$ of the discriminator consist of linear operations (which are constrained by clipping), adding biases (an operation with Lipschitz constant one), and activation functions (taken to be LipSwish). Thus the Lipschitz constant of the overall vector field is at most one, as desired.

### 5.1 Experiments

We test the SDE-GAN on a dataset of time-varying Ornstein–Uhlenbeck samples. For space only a subset of results are shown; see Appendix F for further details of the dataset, optimiser, and so on.

See Table 3 for the results. We see that the test metrics substantially improve with clipping, over gradient penalty (which struggles due to numerical errors in the double adjoint). The lack of double backward additionally implies a computational speed-up. This reduced training time from 55 hours to just 33 hours. Switching to reversible Heun additionally and substantially improves the test metrics, and further reduced training time to 29 hours; a speed improvement of $1.87\times$.

## 6 Discussion

### 6.1 Available implementation in `torchsde`

To facilitate the use of the techniques introduced here – in particular without requiring a technical background in numerical SDEs – we have contributed implementations of both the reversible Heun method and the Brownian Interval to the open-source `torchsde` [42] package. (In which the Brownian Interval has already become the default choice, due to its speed.)

### 6.2 Limitations

The reversible Heun method, Brownian Interval, and training of SDE-GANs via clipping, all appear to be strict improvements over previous techniques. Across our experiments we have observed no limitations relative to previous techniques.

### 6.3 Ethical statement

No significant negative societal impacts are anticipated as a result of this work. A positive environmental impact is anticipated, due to the reduction in compute costs implied by the techniques introduced. See Appendix G for a more in-depth discussion.

## 7 Conclusion

We have introduced several improvements over the previous state-of-the-art for Neural SDEs, with respect to both training speed and test metrics. This has been accomplished through several novel technical innovations, including a first-of-its-kind algebraically reversible SDE solver; a fast, exact, and memory efficient way of sampling and reconstructing Brownian motion; and the development of SDE-GANs via careful clipping and choice of activation function.

## Acknowledgments and Disclosure of Funding

PK was supported by the EPSRC grant EP/L015811/1. PK, JF, TL were supported by the Alan Turing Institute under the EPSRC grant EP/N510129/1. PK thanks Chris Rackauckas for discussions related to the reversible Heun method.

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
