```
1  from torch import randn_like
2  from torch.nn import Module
3  from torchtyping import TensorType
4
5  def euler_maruyama_solve(y0       : TensorType["batch", 1],
6                           dt       : TensorType[()],  # scalar
7                           num_steps: int,
8                           drift    : Module,
9                           diffusion: Module
10                          ) ->      list[TensorType["batch", 1]]:
11     ys = [y0]
12     for _ in range(num_steps):
13         ys.append(ys[-1] + drift(ys[-1]) * dt
14                   + diffusion(ys[-1]) * randn_like(ys[-1]) * dt.sqrt())
15     return ys
```

Figure 4: Euler–Maruyama method in PyTorch. Type annotations (Python 3.9+) are included and `torchtyping` [49] used to indicate shapes of tensors.

## A    RNNs as discretised SDEs

Consider the autonomous one-dimensional Itô SDE

$$\mathrm{d}Y_t = \mu(Y_t)\,\mathrm{d}t + \sigma(Y_t)\,\mathrm{d}W_t,$$

with $Y_t, \mu(Y_t), \sigma(Y_t), W_t \in \mathbb{R}$. Then its numerical Euler–Maruyama discretisation is

$$Y_{n+1} = Y_n + \mu(Y_n)\Delta t + \sigma(Y_n)\Delta W_n,$$

where $\Delta t$ is some fixed time step and $W_n \sim \mathcal{N}(0, \Delta t)$. This may be implemented in very few lines of PyTorch code – see Figure 4 – and subject to a suitable loss function between distributions, such as the KL divergence [15] or Wasserstein distance [16], simply backpropagated through in the usual way.

In this way we see that a discretised SDE is simply an RNN consuming random noise.

**Neural networks as discretised Neural Differential Equations**    In passing we note that this is a common occurrence: many popular neural network architectures may be interpreted as discretised differential equations.

Residual networks are discretisations of ODEs [43, 44]. RNNs are discretised controlled differential equations [45, 46, 47].

StyleGAN2 and denoising diffusion probabilistic models are both essentially discretised SDEs [30, 48].

Many invertible neural networks resemble discretised differential equations; for example using an explicit Euler method on the forward pass, and recovering the intermediate computations via the implicit Euler method on the backward pass [28, 38].

## B    Training criteria for Neural SDEs

**SDEs as GANs**    One classical way of fitting (non-neural) SDEs is to pick some prespecified functions of interest $F_1, \ldots, F_N$, and then ask that $\mathbb{E}_Y[F_i(Y)] \approx \mathbb{E}_{Y_{\text{true}}}[F_i(Y_{\text{true}})]$ for all $i$. For example this may be done by optimising

$$\min_\theta \sum_{i=1}^N \left( \mathbb{E}_Y[F_i(Y)] - \mathbb{E}_{Y_{\text{true}}}[F_i(Y_{\text{true}})] \right)^2.$$

This ensures that the model and the data behave the same with respect to the functions $F_i$. (Known as either 'witness functions' or 'payoff functions' depending on the field.)

Kidger et al. [16] generalise this by replacing $F_1, \ldots F_N$ with a parameterised function $F_\phi$ – a neural network with parameters $\phi$ – and training adversarially:

$$\min_\theta \max_\phi \left( \mathbb{E}_Y \left[ F_\phi(Y) \right] - \mathbb{E}_{Y_{\text{true}}} \left[ F_\phi(Y_{\text{true}}) \right] \right). \tag{9}$$

Thus making a connection to the GAN literature.

In principle $F_\phi$ could be parameterised as any neural network capable of operating on the path-valued $Y$. There is a natural choice: use a Neural CDE [45]. This is a differential equation capable of acting on path-valued inputs. This means letting the discriminator be $F_\phi(Y) = m_\phi \cdot H_T$, where

$$H_0 = \xi_\phi(Y_0), \qquad \mathrm{d}H_t = f_\phi(t, H_t) \, \mathrm{d}t + g_\phi(t, H_t) \circ \mathrm{d}Y_t,$$

for suitable neural networks $\xi_\phi, f_\phi, g_\phi$ and vector $m_\phi$. This is a deterministic function of the generated sample $Y$. Here $\cdot$ denotes a dot product.

Adding regularisation to control the derivative of $F_\phi$ ensures that equation (3) corresponds to the dual formulation of the Wasserstein distance, so that $Y$ is capable of perfectly matching $Y_{\text{true}}$ given enough data, training time, and model capacity [50].

In our experience, this approach produces models with a very high modelling capacity, but which are somewhat involved to train – GANs being notoriously hard to train stably.

**Latent SDEs**    Li et al. [15] have an alternate approach. Let

$$\xi_\phi \colon \mathbb{R}^y \to \mathbb{R}^x \times \mathbb{R}^x, \qquad \nu_\phi \colon [0, T] \times \mathbb{R}^x \times \{[0, T] \to \mathbb{R}^y\} \to \mathbb{R}^x, \tag{10}$$

be (Lipschitz) neural networks[4] parameterised by $\phi$. Let $(m, s) = \xi_\phi(Y_{\text{true},0})$, let $\widehat{V} \sim \mathcal{N}(m, sI_v,)$, and let

$$\widehat{X}_0 = \zeta_\theta(\widehat{V}), \qquad \mathrm{d}\widehat{X}_t = \nu_\phi(t, \widehat{X}_t, Y_{\text{true}}) \, \mathrm{d}t + \sigma_\theta(t, \widehat{X}_t) \circ \mathrm{d}W_t, \qquad \widehat{Y}_t = \ell_\theta(\widehat{X}_t).$$

Note that $\widehat{X}$ is a random variable *over SDEs*, as $Y_{\text{true}}$ is still a random variable.[5]

They show that the KL divergence

$$\mathrm{KL}\left(\widehat{X} \,\big\|\, X\right) = \mathbb{E}_W \int_0^T \frac{1}{2} \left\| (\sigma_\theta(t, \widehat{X}_t))^{-1} (\mu_\theta(t, \widehat{X}_t) - \nu_\phi(t, \widehat{X}_t, Y_{\text{true}})) \right\|_2^2 \mathrm{d}t,$$

and optimise

$$\min_{\theta, \phi} \mathbb{E}_{Y_{\text{true}}} \left[ (\widehat{Y}_0 - Y_{\text{true},0})^2 + \mathrm{KL}\left(\widehat{V} \,\big\|\, V\right) + \mathbb{E}_W \int_0^T (\widehat{Y}_t - Y_{\text{true},t})^2 \, \mathrm{d}t + \mathrm{KL}\left(\widehat{X} \,\big\|\, X\right) \right].$$

This may be derived as an evidence lower-bound (ELBO). The first two terms are simply a VAE for generating $Y_0$, with latent $V$. The third term and fourth term are a VAE for generating $Y$, by autoencoding $Y_{\text{true}}$ to $\widehat{Y}$, and then fitting $X$ to $\widehat{X}$.

In our experience, this produces less expressive models than the SDE-GAN approach; however the model is easier to train, due to the lack of adversarial training.[6]

# C    Adjoints for SDEs

Recall that we wish to backpropagate from our generated sample $Y$ to the parameters $\theta, \phi$. Here we provide a more complete overview of the options for how this may be done.

---

[4] We do not discuss the regularity of elements of $\{[0, T] \to \mathbb{R}^y\}$ as in practice we will have discrete observations; $\nu_\phi$ is thus commonly parameterised as $\nu_\phi(t, \widehat{X}_t, Y_{\text{true}}) = \nu_\phi^1(t, \widehat{X}_t, \nu_\phi^2(Y_{\text{true}}|_{[t,T]}))$, where $\nu_\phi^1$ is for example an MLP, and $\nu_\phi^2$ is an RNN.

[5] So that $\widehat{V}, \widehat{X}, \widehat{Y}$ might be better denoted $\widehat{V}|Y_{\text{true},0}, \widehat{X}|Y_{\text{true}}, \widehat{Y}|Y_{\text{true}}$ to reflect their dependence on $Y_{\text{true}}$; we elide this for simplicity of notation.

[6] For ease of presentation this section features a slight abuse of notation: writing the KL divergence between random variables rather than probability distributions. It is also a slight specialisation of Li et al. [15], who allow losses other than the $L^2$ loss between data and sample.

**Discretise-then-optimise**  One way is to simply backpropagate through the internals of every numerical solver. (Also known as 'discretise-then-optimise'.) However this requires $\mathcal{O}\left(HT\right)$ memory, where $H$ denotes the amount of memory used to evaluate and backpropagate through each neural network once.

**Optimise-then-discretise**  The continuous adjoint method,[7] also known as "optimise-then-discretise", instead exploits the reversibility of a differential equation. This means that intermediate computations such as $X_t$ for $t < T$ are reconstructed from output computations, and do not need to be held in memory.

Recall equations (5) and (6): given some Stratonovich SDE

$$\mathrm{d}Z_t = \mu(t, Z_t)\,\mathrm{d}t + \sigma(t, Z_t) \circ \mathrm{d}W_t \quad \text{for } t \in [0, T],$$

and a loss $L\colon \mathbb{R}^z \to \mathbb{R}$ on its terminal value $Z_T$, then the adjoint process $A_t = \mathrm{d}L(Z_T)/\mathrm{d}Z_t \in \mathbb{R}^z$ is a (strong) solution to

$$\mathrm{d}A_t^i = -A_t^j \frac{\partial \mu^j}{\partial Z^i}(t, Z_t)\,\mathrm{d}t - A_t^j \frac{\partial \sigma^{j,k}}{\partial Z^i}(t, Z_t) \circ \mathrm{d}W_t^k,$$

which in particular uses the same Brownian motion $W$ as on the forward pass. These may be solved backwards-in-time from $t = T$ to $t = 0$, starting from $Z_T = Z_T$ (computed on the forward pass) and $A_T = {}^{\mathrm{d}L(Z_T)}/\mathrm{d}Z_T$. Then $A_0 = {}^{\mathrm{d}L(Z_T)}/Z_0$ is the desired backpropagated gradient.

The main advantage of the continuous adjoint method is that it reduces the memory footprint to only $\mathcal{O}\left(H + T\right)$. $\mathcal{O}\left(H\right)$ to compute each vector-Jacobian product ($A^j \partial \mu^i / \partial Z^j$ and $A^j \partial \sigma^{i,k} / \partial Z^j$), and $\mathcal{O}\left(T\right)$ to hold the batch of training data.

The main disadvantage (unless using the reversible Heun method) is that the two numerical approximations to $Z_t$, computed in the forward and backward passes of equation (5), are different. This means that the $Z_t$ used as an input in equation (6) does not perfectly match what is used in the forward calculation, and the gradients $A_0$ suffer some error as a result. This slows and worsens training. (Often exacerbating an already tricky training procedure, such as the adversarial training of SDE-GANs.)

**Itô versus Stratonovich**  Note that backpropagation through an Itô SDE may be performed by first adding a $-(\sigma^{j,k}\partial\sigma^{i,k}/\partial Z^j)(t, Z_t)/2$ correction term to $\mu$ in equation (5), which converts it to a Stratonovich SDE, and then applying equation (6).

This additional computational cost – including double autodifferentiation to compute derivatives of the correction term – means that we prefer to use Stratonovich SDEs throughout.

## D   Error analysis of the reversible Heun method

### D.1   Notation and definitions

In this section, we present some of the notation, definitions and assumptions used in our error analysis.

Throughout, we use $\|\cdot\|_2$ to denote the standard Euclidean norm on $\mathbb{R}^n$ and $\langle\cdot,\cdot\rangle$ will be the associated inner product with $\langle x, x \rangle = \|x\|_2^2$. We use the same notation to denote norms for tensors.

For $k \geq 1$, a $k$-tensor on $\mathbb{R}^n$ is simply an element of the $n^k$-dimensional space $\mathbb{R}^{n \times \cdots \times n}$ and can be interpreted as a multilinear map into $\mathbb{R}$ with $k$ arguments from $\mathbb{R}^n$. Hence for a $k$-tensor $T$ on $\mathbb{R}^n$ (with $k \geq 2$) and a vector $v \in \mathbb{R}^n$, we shall define $Tv := T(v, \cdots)$ as a $(k-1)$-tensor on $\mathbb{R}^n$.

Therefore, we can define the operator norm of $T$ recursively as

$$\|T\|_{\mathrm{op}} := \begin{cases} \|T\|_2, & \text{if } T \in \mathbb{R}^n, \\ \sup_{\substack{v \in \mathbb{R}^n, \\ \|v\|_2 \leq 1}} \|Tv\|_{\mathrm{op}}, & \text{if } T \text{ is a } k\text{-tensor on } \mathbb{R}^n \text{ with } k \geq 2. \end{cases} \tag{11}$$

---

[7]Frequently abbreviated to simply 'adjoint method' in the modern literature, although this term is ambiguous as it is also used to refer to backpropagation-through-the-solver in other literature [51].

With a slight abuse of notation, we shall write $\|\cdot\|_2$ instead of $\|\cdot\|_{\mathrm{op}}$ for all $k$-tensors.

In addition, we denote the standard tensor product by $\otimes$. So for a $k_1$-tensor $x$ and $k_2$-tensor $y$ on $\mathbb{R}^n$, $x \otimes y$ is a $(k_1 + k_2)$-tensor on $\mathbb{R}^n$. Moreover, it is straightforward to show that $\|x \otimes y\|_2 \le \|x\|_2 \|y\|_2$ and for a $k$-tensor $T$ on $\mathbb{R}^n$, we have that $Tv := T(v_1, \cdots, v_k)$ for all $v = v_1 \otimes \cdots \otimes v_k \in (\mathbb{R}^n)^{\otimes k}$.

We suppose that $(\Omega, \mathcal{F}, \mathbb{P}; \{\mathcal{F}_t\}_{t \ge 0})$ is a filtered probability space carrying a standard $d$-dimensional Brownian motion. We consider the following Stratonovich SDE over the finite time horizon $[0, T]$,

$$\mathrm{d}y_t = f(t, y_t)\, \mathrm{d}t + g(t, y_t) \circ \mathrm{d}W_t, \tag{12}$$

where $y = \{y_t\}_{t \in [0,T]}$ is a continuous $\mathbb{R}^e$-valued stochastic process and $f : [0, T] \times \mathbb{R}^e \to \mathbb{R}^e$, $g : [0, T] \times \mathbb{R}^e \to \mathbb{R}^{e \times d}$ are functions. Without loss of generality, we rewrite (12) as an autonomous SDE by letting $t \mapsto t$ be a coordinate of $y$. This simplifies notation and results in the following SDE:

$$\mathrm{d}y_t = f(y_t)\, \mathrm{d}t + g(y_t) \circ \mathrm{d}W_t, \tag{13}$$

We shall assume that $f, g$ are bounded and twice continuously differentiable with bounded derivatives,

$$\|D^k f\|_\infty := \sup_{x \in \mathbb{R}^e} \|D^k f(x)\|_2 < \infty, \qquad \|D^k g\|_\infty := \sup_{x \in \mathbb{R}^e} \|D^k g(x)\|_2 < \infty, \tag{14}$$

for $k = 0, 1, 2$, where $\|\cdot\|_2$ denotes the Euclidean operator norm on $k$-tensors given by (11).

For $N \ge 1$, we will compute numerical SDE solutions on $[0, T]$ using a constant step size $h = \frac{T}{N}$. That is, numerical solutions are obtained at times $\{t_n\}_{0 \le n \le N}$ with $t_n := nh$ for $n \in \{0, 1, \cdots, N\}$.

For $0 \le n \le N$, we denote increments of Brownian motion by $W_n := W_{t_{n+1}} - W_{t_n} \sim \mathcal{N}(0, \mathrm{I}_d h)$, where $\mathrm{I}_d$ is the $d \times d$ identity matrix. We use $h_{\max} > 0$ to denote an upper bound for the step size $h$.

Given a random vector $X$, taking its values in $\mathbb{R}^n$, we define the $\mathbb{L}_p$ norm of $X$ for $p \ge 1$ as

$$\|X\|_{\mathbb{L}_p} := \mathbb{E}\big[\|X\|_2^p\big]^{\frac{1}{p}}, \tag{15}$$

Similarly, for a stochastic process $\{X_t\}$, the $\mathcal{F}_{t_n}$-conditional $\mathbb{L}_p$ norm of $X_t$ is

$$\|X_t\|_{\mathbb{L}_p^n} := \mathbb{E}_n\big[\|X_t\|_2^p\big]^{\frac{1}{p}} = \mathbb{E}\big[\|X_t\|_2^p \,\big|\, \mathcal{F}_{t_n}\big]^{\frac{1}{p}}, \tag{16}$$

for $t \ge t_n$.

We say that $x(h) = O(h^\gamma)$ if there exists a constant $C > 0$, depending on $h_{\max}$ but not $h$, such that $|x(h)| \le Ch^\gamma$ for $h \in (0, h_{\max}]$. We also use big-$O$ notation for estimating quantities in $\mathbb{L}_p$ and $\mathbb{L}_p^n$.

We use $\|\mathcal{N}(0, \mathrm{I}_d)\|_{\mathbb{L}_p}$ to denote the $\mathbb{L}_p$ norm of a standard $d$-dimensional normal random vector. Thus we have $\|\mathcal{N}(0, \mathrm{I}_d)\|_{\mathbb{L}_2} = \sqrt{d}$, $\|\mathcal{N}(0, \mathrm{I}_d)\|_{\mathbb{L}_4} = (d^2 + 2d)^{\frac{1}{4}}$ and $\|W_n\|_{\mathbb{L}_p} = \|\mathcal{N}(0, \mathrm{I}_d)\|_{\mathbb{L}_p} h^{\frac{1}{2}}$.

We recall the reversible Heun method given by Algorithm 1.

**Definition D.1** (Reversible Heun method). *For $N \ge 1$, we construct a numerical solution $\{Y_n, Z_n\}_{0 \le n \le N}$ for the SDE (13) by setting $Y_0 = Z_0 = y_0$ and, for each $n \in \{0, 1, \cdots, N-1\}$, defining $(Y_{n+1}, Z_{n+1})$ from $(Y_n, Z_n)$ as*

$$Z_{n+1} := 2Y_n - Z_n + f(Z_n)h + g(Z_n)W_n, \tag{17}$$

$$Y_{n+1} := Y_n + \frac{1}{2}\big(f(Z_n) + f(Z_{n+1})\big)h + \frac{1}{2}\big(g(Z_n) + g(Z_{n+1})\big)W_n, \tag{18}$$

*where $W_n := W_{t_{n+1}} - W_{t_n} \sim \mathcal{N}(0, \mathrm{I}_d h)$ is an increment of a $d$-dimensional Brownian motion $W$. The collection $\{Y_n\}_{0 \le n \le N}$ is the numerical approximation to the true solution.*

**Remark D.2.** Whilst it is not used in our analysis, the above numerical method is time-reversible as

$$Z_n = 2Y_{n+1} - Z_{n+1} - f(Z_{n+1})h - g(Z_{n+1})W_n,$$

$$Y_n = Y_{n+1} - \frac{1}{2}\big(f(Z_{n+1}) + f(Z_n)\big)h - \frac{1}{2}\big(g(Z_{n+1}) + g(Z_n)\big)W_n.$$

To simplify notation, we shall define another numerical solution $\{\widetilde{Z}_n\}_{0 \le n \le N}$ with $\widetilde{Z}_n := 2Y_n - Z_n$.

Our analysis is outlined as follows. In Section D.2, we show that when $h$ is sufficiently small, $\|Y_n - Z_n\|_{\mathbb{L}_4} = O(\sqrt{h})$. The choice of $\mathbb{L}_4$ instead of $\mathbb{L}_2$ is important in subsequent error estimates. In Section D.3, we show that the reversible Heun method converges to the Stratonovich solution of SDE (13) in the $\mathbb{L}_2$-norm at a rate of $O(\sqrt{h})$. This matches the convergence rate of standard SDE solvers such as the Heun and midpoint methods. In Section D.4, we consider the case where $g$ is constant (i.e. additive noise) and show that the $\mathbb{L}_2$ convergence rate of our method becomes $O(h)$. We also give numerical evidence that the the reversible Heun method can achieve second order weak convergence for SDEs with additive noise. In Section D.5, we consider the stability of the reversible Heun method in the ODE setting. We show that it has the same absolute stability region for a linear test equation as the (reversible) asynchronous leapfrog integrator proposed for Neural ODEs in [32].

## D.2 Approximation error between components of the reversible Heun method

The key idea underlying our analysis is to consider two steps of the numerical method, which gives,

$$Z_{n+2} := Z_n + 2f(Z_{n+1})h + g(Z_{n+1})(W_n + W_{n+1}), \tag{19}$$

$$
\begin{aligned}
Y_{n+2} := Y_n + \frac{1}{2}\big(f(Z_n) + 2f(Z_{n+1}) + f(Z_{n+2})\big)h \\
+ \frac{1}{2}\big(g(Z_n) + g(Z_{n+1})\big)W_n + \frac{1}{2}\big(g(Z_{n+1}) + g(Z_{n+2})\big)W_{n+1}.
\end{aligned}
\tag{20}
$$

Thus $Z$ is propagated by a midpoint method and $Y$ is propagated by a trapezoidal rule / Heun method. To prove that $\{Z_n\}$ and $\{Y_n\}$ are close together when $h$ is small, we shall use the Taylor expansions:

**Theorem D.3** (Taylor expansions of vector fields). *Let $F$ be a bounded and twice continuously differentiable function on $\mathbb{R}^e$ with bounded derivatives (i.e. we can set $F = f$ or $g$). Then for $n \geq 0$,*

$$F(Z_{n+1}) = F(\widetilde{Z}_n) + F'(\widetilde{Z}_n)(Z_{n+1} - \widetilde{Z}_n) + R_n^{F,1},$$

$$F(Z_{n+2}) = F(Z_n) + F'(Z_n)(Z_{n+2} - Z_n) + R_n^{F,2},$$

*where the remainder terms $R_n^{F,1}$ and $R_n^{F,2}$ satisfy the following estimates for any fixed $p \geq 1$,*

$$\big\|R_n^{F,1}\big\|_{\mathbb{L}_p^n} = O(h),$$

$$\big\|R_n^{F,2}\big\|_{\mathbb{L}_p^n} = O(h).$$

*Proof.* By Taylor's theorem with integral remainder [52, Theorem 3.5.6], the $R_n^{F,i}$ terms are given by

$$R_n^{F,1} = \int_0^1 (1-t)F''\big(\widetilde{Z}_n + t(Z_{n+1} - \widetilde{Z}_n)\big)\, \mathrm{d}t\, \big(Z_{n+1} - \widetilde{Z}_n\big)^{\otimes 2},$$

$$R_n^{F,2} = \int_0^1 (1-t)F''\big(Z_n + t(Z_{n+2} - Z_n)\big)\, \mathrm{d}t\, \big(Z_{n+2} - Z_n\big)^{\otimes 2}.$$

We first note that for $\mathbb{R}^e$-valued random vectors $X_1$ and $X_2$, we have

$$
\begin{aligned}
\left\| \int_0^1 (1-t)F''(X_1 + tX_2)\, \mathrm{d}t\, X_2^{\otimes 2} \right\|_{\mathbb{L}_p^n}
&= \mathbb{E}_n\left[ \left\| \int_0^1 (1-t)F''(X_1 + tX_2)\, \mathrm{d}t\, X_2^{\otimes 2} \right\|_2^p \right]^{\frac{1}{p}} \\
&\leq \mathbb{E}_n\left[ \left\| \int_0^1 (1-t)^{k-1}F''(X_1 + tX_2)\, \mathrm{d}t \right\|_2^p \|X_2^{\otimes 2}\|_2^p \right]^{\frac{1}{p}} \\
&\leq \mathbb{E}_n\left[ \int_0^1 \left\| (1-t)F''(X_1 + tX_2) \right\|_2^p \mathrm{d}t\, \|X_2\|_2^{2p} \right]^{\frac{1}{p}} \\
&\leq (p+1)^{-\frac{1}{p}} \|F''\|_\infty \|X_2\|_{\mathbb{L}_{2p}^n}^2,
\end{aligned}
$$

where the inequality $\left\| \int_0^1 \cdot \, dt \right\|_2^p \le \int_0^1 \| \cdot \|_2^p \, dt$ follows by Jensen's inequality and the convexity of $x \mapsto x^p$. Thus, it is enough to estimate $Z_{n+1} - \widetilde{Z}_n$ and $Z_{n+2} - Z_n$ using Minkowski's inequality.

$$
\begin{aligned}
\left\| Z_{n+1} - \widetilde{Z}_n \right\|_{\mathbb{L}_{2p}^n}^2 &= \left\| f(Z_n)h + g(Z_n)W_n \right\|_{\mathbb{L}_{2p}^n}^2 \\
&\le 2\left\| f(Z_n)h \right\|_{\mathbb{L}_{2p}^n}^2 + 2\left\| g(Z_n)W_n \right\|_{\mathbb{L}_{2p}^n}^2 \\
&\le 2\|f\|_\infty^2 h^2 + 2\|g\|_\infty^2 \|\mathcal{N}(0,\mathrm{I}_d)\|_{\mathbb{L}_{2p}}^2 h,
\end{aligned}
$$

$$
\begin{aligned}
\left\| Z_{n+2} - Z_n \right\|_{\mathbb{L}_{2p}^n}^2 &= \left\| 2f(Z_{n+1})h + g(Z_{n+1})(W_n + W_{n+1}) \right\|_{\mathbb{L}_{2p}^n}^2 \\
&\le 2\left\| 2f(Z_{n+1})h \right\|_{\mathbb{L}_{2p}^n}^2 + 2\left\| g(Z_{n+1})(W_n + W_{n+1}) \right\|_{\mathbb{L}_{2p}^n}^2 \\
&\le 4\|f\|_\infty^2 h^2 + 4\|g\|_\infty^2 \|\mathcal{N}(0,\mathrm{I}_d)\|_{\mathbb{L}_{2p}}^2 h,
\end{aligned}
$$

where we also used the inequality $(a+b)^2 \le 2a^2 + 2b^2$. The result now follows from the above. $\quad\square$

With Theorem D.3, it is straightforward to derive a Taylor expansion for the difference $Y_{n+2} - Z_{n+2}$.

**Theorem D.4.** *For $n \in \{0, 1, \cdots, N-2\}$, the difference $Y_{n+2} - Z_{n+2}$ can be expanded as*

$$
\begin{aligned}
Y_{n+2} - Z_{n+2} = Y_n - Z_n &+ \Big(f(Z_n) - f(\widetilde{Z}_n)\Big)h + \frac{1}{2}\Big(g(Z_n) - g(\widetilde{Z}_n)\Big)(W_n + W_{n+\frac{1}{2}}) \quad (21) \\
&+ \frac{1}{2}\Big(g'(Z_n)\big(g(\widetilde{Z}_n)(W_n + W_{n+1})\big)\Big)W_{n+1} \\
&- \frac{1}{2}\Big(g'(\widetilde{Z}_n)\big(g(Z_n)W_n\big)\Big)(W_n + W_{n+1}) + R_n^Z,
\end{aligned}
$$

*where the remainder term $R_n^Z$ satisfies $\left\| R_n^Z \right\|_{\mathbb{L}_2^n} = O(h^{\frac{3}{2}})$.*

*Proof.* Expanding the components (17), (18) of the reversible Heun method with Theorem D.3 gives

$$
\begin{aligned}
\big(Y_{n+2} &- Z_{n+2}\big) - \big(Y_n - Z_n\big) \\
&= \frac{1}{2}f(Z_n)h + \frac{1}{2}f(Z_{n+2})h - f(Z_{n+1})h \\
&\quad + \frac{1}{2}g(Z_n)W_n + \frac{1}{2}g(Z_{n+2})W_{n+1} - \frac{1}{2}g(Z_{n+1})(W_n + W_{n+1}) \\
&= \frac{1}{2}f(Z_n)h + \frac{1}{2}g(Z_n)W_n + \frac{1}{2}\Big(f(Z_n) + f'(Z_n)(Z_{n+2} - Z_n) + R_n^{f,2}\Big)h \\
&\quad - \Big(f(\widetilde{Z}_n) + f'(\widetilde{Z}_n)(Z_{n+1} - \widetilde{Z}_n) + R_n^{f,1}\Big)h \\
&\quad + \frac{1}{2}\Big(g(Z_n) + g'(Z_n)(Z_{n+2} - Z_n) + R_n^{g,2}\Big)W_{n+1} \\
&\quad - \frac{1}{2}\Big(g(\widetilde{Z}_n) + g'(\widetilde{Z}_n)(Z_{n+1} - \widetilde{Z}_n) + R_n^{g,1}\Big)(W_n + W_{n+1}).
\end{aligned}
$$

Since $R_n^{f,1}, R_n^{f,2}, R_n^{g,1}, R_n^{g,2} \sim O(h)$, which was shown in Theorem D.3, the above simplifies to

$$
\begin{aligned}
\big(Y_{n+2} &- Z_{n+2}\big) - \big(Y_n - Z_n\big) \\
&= \Big(f(Z_n) - f(\widetilde{Z}_n)\Big)h + \frac{1}{2}\Big(g(Z_n) - g(\widetilde{Z}_n)\Big)(W_n + W_{n+\frac{1}{2}}) \\
&\quad + \frac{1}{2}\Big(f'(Z_n)(Z_{n+2} - Z_n)\Big)h - \Big(f'(\widetilde{Z}_n)(Z_{n+1} - \widetilde{Z}_n)\Big)h \\
&\quad + \frac{1}{2}\Big(g'(Z_n)(Z_{n+2} - Z_n)\Big)W_{n+1} - \frac{1}{2}\Big(g'(\widetilde{Z}_n)(Z_{n+1} - \widetilde{Z}_n)\Big)(W_n + W_{n+1}) + O(h^{\frac{3}{2}}).
\end{aligned}
$$

Substituting the formulae (17) and (19) for $Z_{n+1}$ and $Z_{n+2}$ respectively produces

$$
\begin{aligned}
\big(Y_{n+2} &- Z_{n+2}\big) - \big(Y_n - Z_n\big) \\
&= \Big(f(Z_n) - f(\widetilde{Z}_n)\Big)h + \frac{1}{2}\Big(g(Z_n) - g(\widetilde{Z}_n)\Big)\big(W_n + W_{n+\frac{1}{2}}\big) \\
&\quad + \frac{1}{2}\Big(f'(Z_n)\big(2f(Z_{n+1})h + g(Z_{n+1})(W_n + W_{n+1})\big)\Big)h \\
&\quad - \Big(f'(\widetilde{Z}_n)\big(f(Z_n)h + g(Z_n)W_n\big)\Big)h \\
&\quad + \frac{1}{2}\Big(g'(Z_n)\big(2f(Z_{n+1})h + g(Z_{n+1})(W_n + W_{n+1})\big)\Big)W_{n+1} \\
&\quad - \frac{1}{2}\Big(g'(\widetilde{Z}_n)\big(f(Z_n)h + g(Z_n)W_n\big)\Big)(W_n + W_{n+1}) + O\big(h^{\frac{3}{2}}\big).
\end{aligned}
$$

As $f(Y_{n+1})$ and $g(Y_{n+1})$ are bounded by $\|f\|_\infty$ and $\|g\|_\infty$, collecting the $O\big(h^{\frac{3}{2}}\big)$ terms yields

$$
\begin{aligned}
\big(Y_{n+2} - Z_{n+2}\big) - \big(Y_n - Z_n\big) &= \Big(f(Z_n) - f(\widetilde{Z}_n)\Big)h + \frac{1}{2}\Big(g(Z_n) - g(\widetilde{Z}_n)\Big)\big(W_n + W_{n+\frac{1}{2}}\big) \\
&\quad + \frac{1}{2}\Big(g'(Z_n)\big(g(Z_{n+1})(W_n + W_{n+1})\big)\Big)W_{n+1} \\
&\quad - \frac{1}{2}\Big(g'(\widetilde{Z}_n)\big(g(Z_n)W_n\big)\Big)(W_n + W_{n+1}) + O\big(h^{\frac{3}{2}}\big).
\end{aligned}
$$

We note that it is direct consequence of Theorem D.3 that $g(Z_{n+1}) = g(\widetilde{Z}_n) + O(\sqrt{h})$. Thus,

$$
\begin{aligned}
\big(Y_{n+2} - Z_{n+2}\big) - \big(Y_n - Z_n\big) &= \Big(f(Z_n) - f(\widetilde{Z}_n)\Big)h + \frac{1}{2}\Big(g(Z_n) - g(\widetilde{Z}_n)\Big)\big(W_n + W_{n+\frac{1}{2}}\big) \\
&\quad + \frac{1}{2}\Big(g'(Z_n)\big(g(\widetilde{Z}_n)(W_n + W_{n+1})\big)\Big)W_{n+1} \\
&\quad - \frac{1}{2}\Big(g'(\widetilde{Z}_n)\big(g(Z_n)W_n\big)\Big)(W_n + W_{n+1}) + O\big(h^{\frac{3}{2}}\big),
\end{aligned}
$$

which gives the desired result. $\qquad\square$

Using the expansion (21), we shall derive estimates for the $\mathbb{L}_4$-norm of the difference $Y_{n+2} - Z_{n+2}$.

**Theorem D.5** (Local bound for $Y - Z$). *Let $h_{\max} > 0$ be fixed. Then there exist constants $c_1, c_2 > 0$ such that for $n \in \{0, 1, \cdots, N - 2\}$,*

$$
\big\|Y_{n+2} - Z_{n+2}\big\|_{\mathbb{L}_4^n}^4 \leq e^{c_1 h}\big\|Y_n - Z_n\big\|_2^4 + c_2 h^3,
$$

*provided $h \leq h_{\max}$.*

*Proof.* To begin, we will expand $\big\|Y_{n+2} - Z_{n+2}\big\|_{\mathbb{L}_4^n}^4$ as

$$
\begin{aligned}
\big\|Y_{n+2} &- Z_{n+2}\big\|_{\mathbb{L}_4^n}^4 \\
&= \mathbb{E}_n\Big[\Big(\|Y_n - Z_n\|_2^2 + 2\big\langle Y_n - Z_n, (Y_{n+2} - Z_{n+2}) - (Y_n - Z_n)\big\rangle \\
&\qquad + \big\|(Y_{n+2} - Z_{n+2}) - (Y_n - Z_n)\big\|_2^2\Big)^2\Big] \\
&= \|Y_n - Z_n\|_2^4 + 4\,\mathbb{E}_n\Big[\|Y_n - Z_n\|_2^2\langle Y_n - Z_n, (Y_{n+2} - Z_{n+2}) - (Y_n - Z_n)\rangle\Big] \\
&\quad + 4\,\mathbb{E}_n\Big[\langle Y_n - Z_n, (Y_{n+2} - Z_{n+2}) - (Y_n - Z_n)\rangle^2\Big] \\
&\quad + 4\,\mathbb{E}_n\Big[\langle Y_n - Z_n, (Y_{n+2} - Z_{n+2}) - (Y_n - Z_n)\rangle\big\|(Y_{n+2} - Z_{n+2}) - (Y_n - Z_n)\big\|_2^2\Big] \\
&\quad + 2\,\mathbb{E}_n\Big[\|Y_n - Z_n\|_2^2\big\|(Y_{n+2} - Z_{n+2}) - (Y_n - Z_n)\big\|_2^2\Big] \\
&\quad + \big\|(Y_{n+2} - Z_{n+2}) - (Y_n - Z_n)\big\|_{\mathbb{L}_4^n}^4.
\end{aligned}
$$

Using the Cauchy-Schwarz inequality along with the fact that $\|\cdot\|_{\mathbb{L}_2^n} \leq \|\cdot\|_{\mathbb{L}_4^n}$, we have

$$
\begin{aligned}
&\left\|Y_{n+2} - Z_{n+2}\right\|_{\mathbb{L}_4^n}^4 \\
&\quad \leq \left\|Y_n - Z_n\right\|_2^4 + 4\left\|Y_n - Z_n\right\|_2^2 \left\langle Y_n - Z_n, \mathbb{E}_n\left[\left(Y_{n+2} - Z_{n+2}\right) - \left(Y_n - Z_n\right)\right]\right\rangle \qquad (22) \\
&\qquad + 6\left\|Y_n - Z_n\right\|_2^2 \left\|\left(Y_{n+2} - Z_{n+2}\right) - \left(Y_n - Z_n\right)\right\|_{\mathbb{L}_2^n}^2 \\
&\qquad + 4\left\|Y_n - Z_n\right\|_2 \left\|\left(Y_{n+2} - Z_{n+2}\right) - \left(Y_n - Z_n\right)\right\|_{\mathbb{L}_4^n}^3 \\
&\qquad + \left\|\left(Y_{n+2} - Z_{n+2}\right) - \left(Y_n - Z_n\right)\right\|_{\mathbb{L}_4^n}^4.
\end{aligned}
$$

By Theorem D.4, the second term can be expanded as

$$
\begin{aligned}
&2\,\mathbb{E}_n\left[\left\langle Y_n - Z_n, \left(Y_{n+2} - Z_{n+2}\right) - \left(Y_n - Z_n\right)\right\rangle\right] \\
&\quad = 2\left\langle Y_n - Z_n, \mathbb{E}_n\left[\left(Y_{n+2} - Z_{n+2}\right) - \left(Y_n - Z_n\right)\right]\right\rangle \\
&\quad = 2\left\langle Y_n - Z_n, \mathbb{E}_n\left[\left(f(Z_n) - f(\widetilde{Z}_n)\right)h + \frac{1}{2}\left(g(Z_n) - g(\widetilde{Z}_n)\right)\left(W_n + W_{n+\frac{1}{2}}\right)\right]\right\rangle \\
&\qquad + 2\left\langle Y_n - Z_n, \frac{1}{2}\mathbb{E}_n\left[\left(g'(Z_n)\left(g(\widetilde{Z}_n)(W_n + W_{n+1})\right)\right)W_{n+1}\right]\right\rangle \\
&\qquad - 2\left\langle Y_n - Z_n, \frac{1}{2}\mathbb{E}_n\left[\left(g'(\widetilde{Z}_n)\left(g(Z_n)W_n\right)\right)(W_n + W_{n+1}) + R_n^Z\right]\right\rangle \\
&\quad = 2\left\langle Y_n - Z_n, \left(f(Z_n) - f(\widetilde{Z}_n)\right)\right\rangle h + \left\langle Y_n - Z_n, \left(g'(Z_n)g(\widetilde{Z}_n) - g'(Z_n)g(Z_n)\right)\mathrm{I}_d\right\rangle h \\
&\qquad + \left\langle Y_n - Z_n, \left(g'(Z_n)g(Z_n) - g'(\widetilde{Z}_n)g(Z_n)\right)\mathrm{I}_d\right\rangle h + 2\left\langle Y_n - Z_n, \mathbb{E}_n\left[R_n^Z\right]\right\rangle.
\end{aligned}
$$

The above can be estimated using the Cauchy-Schartz inequality and Young's inequality as

$$
\begin{aligned}
&\left|2\,\mathbb{E}_n\left[\left\langle Y_n - Z_n, \left(Y_{n+2} - Z_{n+2}\right) - \left(Y_n - Z_n\right)\right\rangle\right]\right| \\
&\quad \leq 2\left\|Y_n - Z_n\right\|_2 \left\|f(Z_n) - f(\widetilde{Z}_n)\right\|_2 h + \left\|Y_n - Z_n\right\|_2 \left\|\left(g'(Z_n)g(\widetilde{Z}_n) - g'(Z_n)g(Z_n)\right)\mathrm{I}_d\right\|_2 h \\
&\qquad + \left\|Y_n - Z_n\right\|_2 \left\|\left(g'(Z_n)g(Z_n) - g'(\widetilde{Z}_n)g(Z_n)\right)\mathrm{I}_d\right\|_2 h + 2\left\|Y_n - Z_n\right\|_2 \left\|\mathbb{E}_n\left[R_n^Z\right]\right\|_2 \\
&\quad \leq 2\|f'\|_\infty \left\|Y_n - Z_n\right\|_2 \left\|Z_n - \widetilde{Z}_n\right\|_2 h + \|g'\|_\infty^2 \left\|Y_n - Z_n\right\|_2 \left\|Z_n - \widetilde{Z}_n\right\|_2 dh \\
&\qquad + \|g''\|_\infty \|g\|_\infty \left\|Y_n - Z_n\right\|_2 \left\|Z_n - \widetilde{Z}_n\right\|_2 dh + 2\left\|Y_n - Z_n\right\|_2 h^{\frac{1}{2}} \left\|\mathbb{E}_n\left[R_n^Z\right]\right\|_2 h^{-\frac{1}{2}} \\
&\quad \leq \left(4\|f'\|_\infty + 2\|g'\|_\infty^2 d + 2\|g''\|_\infty \|g\|_\infty d + 1\right)\left\|Y_n - Z_n\right\|_2^2 h + \frac{1}{2}\left\|\mathbb{E}_n\left[R_n^Z\right]\right\|_2^2 h^{-1}.
\end{aligned}
$$

By Jensen's inequality and Theorem D.4, we have that $\left\|\mathbb{E}_n\left[R_n^Z\right]\right\|_2^2 \leq \left\|R_n^Z\right\|_{\mathbb{L}_2^n}^2 = O(h^3)$. Therefore

$$
\left|\mathbb{E}_n\left[\left\langle Y_n - Z_n, \left(Y_{n+2} - Z_{n+2}\right) - \left(Y_n - Z_n\right)\right\rangle\right]\right| \leq O(h) \cdot \left\|Y_n - Z_n\right\|_2^2 + O(h^2).
$$

The final term in (22) can be estimated using Minkowski's inequality as

$$
\begin{aligned}
&\left\|\left(Y_{n+2} - Z_{n+2}\right) - \left(Y_n - Z_n\right)\right\|_{\mathbb{L}_p^n} \\
&\quad = \left\|\left(f(Z_n) - f(\widetilde{Z}_n)\right)h + \frac{1}{2}\left(g(Z_n) - g(\widetilde{Z}_n)\right)\left(W_n + W_{n+\frac{1}{2}}\right)\right. \\
&\qquad + \frac{1}{2}\left(g'(Z_n)\left(g(\widetilde{Z}_n)(W_n + W_{n+1})\right)\right)W_{n+1} \\
&\qquad \left.- \frac{1}{2}\left(g'(\widetilde{Z}_n)\left(g(Z_n)W_n\right)\right)(W_n + W_{n+1}) + R_n^Z\right\|_{\mathbb{L}_p^n} \\
&\quad \leq O(\sqrt{h}) \cdot \left\|Y_n - Z_n\right\|_2 + O(h).
\end{aligned}
$$

Since $(a+b)^k \leq 2^{k-1}(a^k + b^k)$ for $a, b \geq 0$ (which follows from Jensen's inequality), this gives

$$\left\|\left(Y_{n+2} - Z_{n+2}\right) - \left(Y_n - Z_n\right)\right\|_{\mathbb{L}_p^n}^k \leq O\left(h^{\frac{1}{2}k}\right) \cdot \left\|Y_n - Z_n\right\|_2^k + O\left(h^k\right).$$

Hence the estimate (22) becomes

$$\left\|Y_{n+2} - Z_{n+2}\right\|_{\mathbb{L}_4^n}^4 \leq \left\|Y_n - Z_n\right\|_2^4 + 4\left\|Y_n - Z_n\right\|_2^2\left(O(h) \cdot \left\|Y_n - Z_n\right\|_2^2 + O\left(h^2\right)\right) \qquad (23)$$
$$+ 6\left\|Y_n - Z_n\right\|_2^2\left(O(h) \cdot \left\|Y_n - Z_n\right\|_2^2 + O\left(h^2\right)\right)$$
$$+ 4\left\|Y_n - Z_n\right\|_2\left(O\left(h^{\frac{3}{2}}\right) \cdot \left\|Y_n - Z_n\right\|_2^3 + O\left(h^3\right)\right)$$
$$+ O\left(h^2\right) \cdot \left\|Y_n - Z_n\right\|_2^4 + O\left(h^4\right).$$

By Young's inequality, we can further estimate the terms which do not contain $\left\|Y_n - Z_n\right\|_2^4$ as

$$\left\|Y_n - Z_n\right\|_2^2 \cdot O\left(h^2\right) = \left\|Y_n - Z_n\right\|_2^2 h^{\frac{1}{2}} \cdot O\left(h^{\frac{3}{2}}\right)$$
$$\leq \frac{1}{2}\left\|Y_n - Z_n\right\|_2^4 h + O\left(h^3\right),$$

$$\left\|Y_n - Z_n\right\|_2 \cdot O\left(h^3\right) = \left\|Y_n - Z_n\right\|_2 h \cdot O\left(h^2\right)$$
$$\leq \frac{1}{2}\left\|Y_n - Z_n\right\|_2^2 h^2 + O\left(h^4\right)$$
$$\leq \frac{1}{4}\left\|Y_n - Z_n\right\|_2^4 h + h^3 + O\left(h^4\right).$$

Finally, by applying the above two estimates to the inequality (23), we arrive at

$$\left\|Y_{n+2} - Z_{n+2}\right\|_{\mathbb{L}_4^n}^4 \leq \left(1 + O(h)\right)\left\|Y_n - Z_n\right\|_{\mathbb{L}_4^n}^4 + O\left(h^3\right),$$

which gives the desired result. $\qquad \square$

We can now prove that $Y$ and $Z$ become close to each other at a rate of $O\left(\sqrt{h}\right)$ in the $\mathbb{L}_4$-norm.

**Theorem D.6** (Global bound for $Y - Z$). *Let $h_{\max} > 0$ be fixed. There exist a constant $C_1 > 0$ such that for all $n \in \{0, 1, \cdots, N\}$,*

$$\left\|Y_n - Z_n\right\|_{\mathbb{L}_4} \leq C_1\sqrt{h},$$

*provided $h \leq h_{\max}$.*

*Proof.* By the tower property of expectations, it follows from Theorem D.5 that

$$\left\|Y_{n+2} - Z_{n+2}\right\|_{\mathbb{L}_4}^4 \leq e^{c_1 h}\left\|Y_n - Z_n\right\|_{\mathbb{L}_4}^4 + c_2 h^3,$$

which, along with the fact that $Y_0 = Z_0$, implies that

$$\left\|Y_{2n} - Z_{2n}\right\|_{\mathbb{L}_4}^4 \leq c_2 \sum_{k=0}^{n-1} e^{c_1 kh} h^2 = c_2 \frac{e^{c_1 nh} - 1}{e^{c_1 h} - 1} h^2 \leq c_2 \frac{e^{\frac{1}{2}c_1 T} - 1}{e^{c_1 h} - 1} h^3 = O\left(h^2\right).$$

It is now straightforward to estimate the $\mathbb{L}_4$-norm of $Y_{2n+1} - Z_{2n+1}$ as

$$\left\|Y_{2n+1} - Z_{2n+1}\right\|_{\mathbb{L}_4}$$
$$\leq \left\|Y_{2n} - Z_{2n}\right\|_{\mathbb{L}_4} + \left\|\left(Y_{2n+1} - Z_{2n+1}\right) - \left(Y_{2n} - Z_{2n}\right)\right\|_{\mathbb{L}_4}$$
$$\leq 2\left\|Y_{2n} - Z_{2n}\right\|_{\mathbb{L}_4} + \frac{1}{2}\left\|\left(f\left(Z_{2n+1}\right) - f\left(Z_{2n}\right)\right)h + \left(g\left(Z_{2n+1}\right) - g\left(Z_{2n}\right)\right)W_{2n}\right\|_{\mathbb{L}_4}$$
$$= O\left(\sqrt{h}\right),$$

which gives the desired result. $\qquad \square$

### D.3 Strong convergence of the reversible Heun method

To begin, we will Taylor expand the terms in the $Y_n \mapsto Y_{n+1}$ update of the reversible Heun method.

**Theorem D.7** (Further Taylor expansions for the vector fields). *Let $F$ be a bounded and twice continuously differentiable function on $\mathbb{R}^e$ with bounded derivatives. Then for $n \geq 0$,*

$$F(Z_n) = F(Y_n) + F'(Y_n)(Z_n - Y_n) + R_n^{F,3},$$

$$F(Z_{n+1}) = F(Y_n) + F'(Y_n)(Y_n - Z_n) + F'(Y_n)(g(Y_n)W_n) + R_n^{F,4},$$

*where the remainder terms $R_n^{F,3}$ and $R_n^{F,4}$ satisfy the following estimates,*

$$\left\| R_n^{F,3} \right\|_{\mathbb{L}_2} = O(h),$$
$$\left\| R_n^{F,4} \right\|_{\mathbb{L}_2} = O(h),$$
$$\left\| R_n^{F,3} W_n \right\|_{\mathbb{L}_2} = O(h^{\frac{3}{2}}),$$
$$\left\| R_n^{F,4} W_n \right\|_{\mathbb{L}_2} = O(h^{\frac{3}{2}}).$$

*Proof.* By Taylor's theorem with integral remainder [52, Theorem 3.5.6], we have that for $k \in \{0, 1\}$,

$$F(Z_{n+k}) = F(Y_n) + F'(Y_n)(Z_{n+k} - Y_n)$$
$$+ \int_0^1 (1 - t) F''(Y_n + t(Z_{n+k} - Y_n)) \, dt \, (Z_{n+k} - Y_n)^{\otimes 2}.$$

By the same argument used in the proof of Theorem D.3, we can estimate the remainder term as

$$\left\| \int_0^1 (1 - t) F''(Y_n + t(Z_{n+k} - Y_n)) \, dt \, (Z_{n+k} - Y_n)^{\otimes 2} \right\|_{\mathbb{L}_2^n}^2 \leq \frac{1}{3} \| F'' \|_\infty^2 \| Z_{n+k} - Y_n \|_{\mathbb{L}_4^n}^4,$$

Using the inequality $(a + b)^4 \leq (2a^2 + 2b^2)^2 \leq 8a^4 + 8b^4$, we have

$$\left\| Z_n - Y_n \right\|_{\mathbb{L}_4^n}^4 = \left\| Y_n - Z_n \right\|_2^4,$$

$$\left\| Z_{n+1} - Y_n \right\|_{\mathbb{L}_4^n}^4 = \left\| Y_n - Z_n + f(Z_n)h + g(Z_n)W_n \right\|_{\mathbb{L}_4^n}^4$$
$$\leq 8 \left\| Y_n - Z_n \right\|_2^4 + 8 \left\| f(Z_n)h + g(Z_n)W_n \right\|_{\mathbb{L}_4^n}^4$$
$$\leq 8 \left\| Y_n - Z_n \right\|_2^4 + 64 \left\| f(Z_n)h \right\|_{\mathbb{L}_4^n}^4 + 64 \left\| g(Z_n)W_n \right\|_{\mathbb{L}_4^n}^2$$
$$\leq 8 \left\| Y_n - Z_n \right\|_2^4 + 64 \| f \|_\infty^4 h^4 + 64 \| g \|_\infty^4 \| \mathcal{N}(0, \mathrm{I}_d) \|_{\mathbb{L}_4}^4 h^2.$$

Therefore, by the tower property of expectations, for $k \in \{0, 1\}$,

$$\left\| \int_0^1 (1 - t) F''(Y_n + t(Z_{n+k} - Y_n)) \, dt \, (Z_{n+k} - Y_n)^{\otimes 2} \right\|_{\mathbb{L}_2}^2 \leq \frac{1}{3} \| F'' \|_\infty^2 \mathbb{E}\left[ \left\| Z_{n+k} - Y_n \right\|_{\mathbb{L}_4^n}^4 \right]$$
$$= O\left( \left\| Y_n - Z_n \right\|_{\mathbb{L}_4}^4 + h^2 \right)$$
$$= O(h^2),$$

by the $O(\sqrt{h})$ global bound on $\| Y_n - Z_n \|_{\mathbb{L}_4}$ in Theorem D.6.

Similarly, for $k \in \{0, 1\}$,

$$\left\| \int_0^1 (1-t) \, F''\big(Y_n + t\big(Z_{n+k} - Y_n\big)\big) \, \mathrm{d}t \, \big(Z_{n+k} - Y_n\big)^{\otimes 2} \, W_n \right\|_{\mathbb{L}_2}^2$$

$$= \mathbb{E}\left[ \left\| \int_0^1 (1-t) \, F''\big(Y_n + t\big(Z_{n+k} - Y_n\big)\big) \, \mathrm{d}t \, \big(Z_{n+k} - Y_n\big)^{\otimes 2} \, W_n \right\|_{\mathbb{L}_2^n}^2 \right]$$

$$\leq \mathbb{E}\left[ \left\| \int_0^1 (1-t) \, F''\big(Y_n + t\big(Z_{n+k} - Y_n\big)\big) \, \mathrm{d}t \right\|_2^2 \left\| \big(Z_{n+k} - Y_n\big)^{\otimes 2} \right\|_2^2 \left\| W_n \right\|_2^2 \right]$$

$$\leq \mathbb{E}\left[ \int_0^1 \left\| (1-t) \, F''\big(Y_n + t\big(Z_{n+k} - Y_n\big)\big) \right\|_2^2 \mathrm{d}t \, \left\| Z_{n+k} - Y_n \right\|_2^4 \left\| W_n \right\|_2^2 \right]$$

$$\leq \frac{1}{3} \|F''\|_\infty^2 \, \mathbb{E}\left[ \left\| Z_{n+k} - Y_n \right\|_2^4 \left\| W_n \right\|_2^2 \right].$$

We consider the $k = 0$ and $k = 1$ cases separately. If $k = 0$, then by the tower property, we have

$$\left\| \int_0^1 (1-t) F''\big(Y_n + t\big(Z_n - Y_n\big)\big) \, \mathrm{d}t \, \big(Z_n - Y_n\big)^{\otimes 2} \, W_n \right\|_{\mathbb{L}_2}^2$$

$$\leq \frac{1}{3} \|F''\|_\infty^2 \, \mathbb{E}\left[ \mathbb{E}_n\left[ \left\| Z_n - Y_n \right\|_2^4 \left\| W_n \right\|_2^2 \right] \right]$$

$$= \frac{1}{3} \|F''\|_\infty^2 \, \mathbb{E}\left[ \left\| Z_n - Y_n \right\|_2^4 \mathbb{E}_n\left[ \left\| W_n \right\|_2^2 \right] \right]$$

$$= O\big(h^3\big).$$

Slightly more care should be taken when $k = 1$ since $Z_{n+1} - Y_n$ is not independent of $W_n$.

$$\left\| \int_0^1 (1-t) \, F''\big(Y_n + t\big(Z_{n+1} - Y_n\big)\big) \, \mathrm{d}t \, \big(Z_{n+1} - Y_n\big)^{\otimes 2} \, W_n \right\|_{\mathbb{L}_2}^2$$

$$\leq \frac{1}{3} \|F''\|_\infty^2 \, \mathbb{E}\left[ \mathbb{E}_n\left[ \left\| Y_n - Z_n + f\big(Z_n\big)h + g\big(Z_n\big)W_n \right\|_2^4 \left\| W_n \right\|_2^2 \right] \right]$$

$$\leq \frac{1}{3} \|F''\|_\infty^2 \, \mathbb{E}\left[ \mathbb{E}_n\left[ 8\left\| Y_n - Z_n \right\|_2^4 \left\| W_n \right\|_2^2 + 8\left\| f\big(Z_n\big)h + g\big(Z_n\big)W_n \right\|_2^4 \left\| W_n \right\|_2^2 \right] \right]$$

$$\leq \frac{8}{3} \|F''\|_\infty^2 \, \mathbb{E}\left[ \left\| Y_n - Z_n \right\|_2^4 \mathbb{E}_n\left[ \left\| W_n \right\|_2^2 \right] + \mathbb{E}_n\left[ \left\| f\big(Z_n\big)h + g\big(Z_n\big)W_n \right\|_2^4 \left\| W_n \right\|_2^2 \right] \right]$$

$$= O\big(h^3\big).$$

Finally, by Theorem D.6 and the Lipschitz continuity of $g$, $F'\big(Y_n\big)\big(Z_{n+1} - Y_n\big)$ can be expanded as

$$F'\big(Y_n\big)\big(Z_{n+1} - Y_n\big)$$
$$= F'\big(Y_n\big)\big(Y_n - Z_n\big) + F'\big(Y_n\big)\big(g\big(Y_n\big)W_n\big) + F'\big(Y_n\big)\big(f\big(Z_n\big)h + \big(g\big(Y_n\big) - g\big(Z_n\big)\big)W_n\big)$$
$$= F'\big(Y_n\big)\big(Y_n - Z_n\big) + F'\big(Y_n\big)\big(g\big(Y_n\big)W_n\big) + O(h).$$

The result follows from the above estimates. $\qquad\square$

We are now in a position to compute the local Taylor expansion of the numerical approximation $Y$.

**Theorem D.8** (Taylor expansion of the reversible Heun method). *For $n \in \{0, 1, \cdots, N-1\}$,*

$$Y_{n+1} = Y_n + f\big(Y_n\big)h + g\big(Y_n\big)W_n + \frac{1}{2}g'\big(Y_n\big)\big(g\big(Y_n\big)W_n\big)W_n + R_n^Y,$$

*where the remainder term satisfies* $\|R_n^Y\|_{\mathbb{L}_2} = O\big(h^{\frac{3}{2}}\big)$.

*Proof.* By Theorem D.7, we can expand $Y_{n+1}$ as

$$Y_{n+1} = Y_n + \frac{1}{2}\big(f(Z_n) + f(Z_{n+1})\big)h + \frac{1}{2}\big(g(Z_n) + g(Z_{n+1})\big)W_n$$

$$= Y_n + \frac{1}{2}\big(f(Y_n) + f'(Y_n)(Z_n - Y_n) + R_n^{f,3}\big)h$$

$$+ \frac{1}{2}\big(f(Y_n) + f'(Y_n)(Y_n - Z_n) + f'(Y_n)(g(Y_n)W_n) + R_n^{f,4}\big)h$$

$$+ \frac{1}{2}\big(g(Y_n) + g'(Y_n)(Z_n - Y_n) + R_n^{g,3}\big)W_n$$

$$+ \frac{1}{2}\big(g(Y_n) + g'(Y_n)(Y_n - Z_n) + g'(Y_n)(g(Y_n)W_n) + R_n^{g,4}\big)W_n$$

$$= Y_n + f(Y_n)h + g(Y_n)W_n + \frac{1}{2}g'(Y_n)\big(g(Y_n)W_n\big)W_n$$

$$+ \frac{1}{2}f'(Y_n)\big(g(Y_n)W_n\big)h + \frac{1}{2}\big(R_n^{f,3} + R_n^{f,4}\big)h + \frac{1}{2}\big(R_n^{g,3} + R_n^{g,4}\big)W_n\,.$$

The result now follows as the bottom line is clearly $O\big(h^{\frac{3}{2}}\big)$ by Theorem D.7. $\qquad\square$

Just as in the numerical analysis of ODE solvers, we also compute Taylor expansions for the solution. In our setting, we consider the following stochastic Taylor expansion for the Stratonovich SDE (13).

**Theorem D.9** (Stratonovich–Taylor expansion, [53, Proposition 5.10.1]). *For $n \in \{0, 1, \cdots, N-1\}$,*

$$y_{(n+1)h} = y_{nh} + f(y_{nh})h + g(y_{nh})W_n + g'(y_{nh})g(y_{nh})\mathbb{W}_n + R_n^y\,,$$

*where $\mathbb{W}_n$ denotes the second iterated integral of Brownian motion, that is the $d \times d$ matrix given by*

$$\mathbb{W}_n := \int_{nh}^{(n+1)h} \big(W_t - W_{nh}\big) \otimes \circ\, \mathrm{d}W_t\,,$$

*and the remainder term satisfies $\|R_n^y\|_{\mathbb{L}_2} = O\big(h^{\frac{3}{2}}\big)$.*

Using the above theorems, we can now obtain a Taylor expansion for the difference $Y_{n+1} - y_{(n+1)h}$.

**Theorem D.10.** *For $n \in \{0, 1, \cdots, N-1\}$, , the difference $Y_{n+2} - Z_{n+2}$ can be expanded as*

$$Y_{n+1} - y_{(n+1)h} = Y_n - y_{nh} + \big(f(Y_n) - f(y_{nh})\big)h + \big(g(Y_n) - g(y_{nh})\big)W_n$$

$$+ \frac{1}{2}\Big(g'(Y_n)g(Y_n) - g'(y_{nh})g(y_{nh})\Big)W_n^{\otimes 2}$$

$$- g'(y_{nh})g(y_{nh})\Big(\mathbb{W}_n - \frac{1}{2}W_n^{\otimes 2}\Big) + R_n\,,$$

*where the remainder term satisfies $\|R_n\|_{\mathbb{L}_2} = O\big(h^{\frac{3}{2}}\big)$.*

*Proof.* Expanding $Y_{n+1} - y_{(n+1)h}$ using Theorem D.8 and Theorem D.9 gives

$$Y_{n+1} - y_{(n+1)h} = Y_{n+1} - y_{nh} - f(y_{nh})h - g(y_{nh})W_n - g'(y_{nh})g(y_{nh})\mathbb{W}_n - R_n^y$$

$$= Y_n - y_{nh} + f(Y_n)h + g(Y_n)W_n + \frac{1}{2}g'(Y_n)\big(g(Y_n)W_n\big)W_n$$

$$- f(y_{nh})h - g(y_{nh})W_n - g'(y_{nh})g(y_{nh})\mathbb{W}_n + R_n^Y - R_n^y\,,$$

where the remainder term $R_n$ satisfies $\|R_n\|_{\mathbb{L}_2} \leq \|R_n^Y\|_{\mathbb{L}_2} + \|R_n^y\|_{\mathbb{L}_2} = O\big(h^{\frac{3}{2}}\big)$. $\qquad\square$

Having derived Taylor expansions for the approximation and solution processes, we will establish the main results of the section (namely, local and global error estimates for the reversible Heun method).

**Theorem D.11** (Local error estimate for the reversible Heun method). *Let $h_{\max} > 0$ be fixed. Then there exist constants $c_3, c_4 > 0$ such that for all $n \in \{0, 1, \cdots, N-1\}$,*

$$\big\|Y_{n+1} - y_{(n+1)h}\big\|_{\mathbb{L}_2}^2 \leq e^{c_3 h}\big\|Y_n - y_{nh}\big\|_{\mathbb{L}_2}^2 + c_4 h^2, \tag{24}$$

*provided $h \leq h_{\max}$.*

*Proof.* Expanding the left hand side of (24) and applying the tower property of expectations yields

$$\left\|Y_{n+1} - y_{(n+1)h}\right\|_{\mathbb{L}_2}^2 = \left\|Y_n - y_{nh}\right\|_{\mathbb{L}_2}^2 + 2\,\mathbb{E}\left[\left\langle Y_n - y_{nh}\,,\left(Y_{n+1} - y_{(n+1)h}\right) - \left(Y_n - y_{nh}\right)\right\rangle\right]$$
$$+ \left\|\left(Y_{n+1} - y_{(n+1)h}\right) - \left(Y_n - y_{nh}\right)\right\|_{\mathbb{L}_2}^2$$
$$= \left\|Y_n - y_{nh}\right\|_{\mathbb{L}_2}^2 + 2\,\mathbb{E}\left[\left\langle Y_n - y_{nh}\,,\mathbb{E}_n\left[\left(Y_{n+1} - y_{(n+1)h}\right) - \left(Y_n - y_{nh}\right)\right]\right\rangle\right]$$
$$+ \left\|\left(Y_{n+1} - y_{(n+1)h}\right) - \left(Y_n - y_{nh}\right)\right\|_{\mathbb{L}_2}^2.$$

A simple application of the Cauchy-Schwarz inequality then gives

$$\left\|Y_{n+1} - y_{(n+1)h}\right\|_{\mathbb{L}_2}^2 \leq \left\|Y_n - y_{nh}\right\|_{\mathbb{L}_2}^2 + \left\|\left(Y_{n+1} - y_{(n+1)h}\right) - \left(Y_n - y_{nh}\right)\right\|_{\mathbb{L}_2}^2 \qquad (25)$$
$$+ 2\,\mathbb{E}\left[\left\|Y_n - y_{nh}\right\|_2\left\|\mathbb{E}_n\left[\left(Y_{n+1} - y_{(n+1)h}\right) - \left(Y_n - y_{nh}\right)\right]\right\|_2\right].$$

To further estimate the above, we note that $\mathbb{W}_n$ and $\frac{1}{2}W_n^{\otimes 2}$ have the same expectation as

$$\mathbb{E}_n\left[\mathbb{W}_n\right] = \mathbb{E}_n\left[\int_{nh}^{(n+1)h}\left(W_t - W_{nh}\right)\otimes\circ\,\mathrm{d}W_t\right]$$
$$= \mathbb{E}_n\left[\int_{nh}^{(n+1)h}\left(W_t - W_{nh}\right)\otimes\mathrm{d}W_t + \frac{1}{2}\mathrm{I}_d h\right]$$
$$= \frac{1}{2}\mathrm{I}_d h,$$

where the second lines follws by the Itô–Stratonovich correction.

This gives the required $O(h)$ cancellation when we expand the final term in (25) using Theorem D.10.

$$\left\|\mathbb{E}_n\left[\left(Y_{n+1} - y_{(n+1)h}\right) - \left(Y_n - y_{nh}\right)\right]\right\|_2$$
$$= \left\|\left(f(Y_n) - f(y_{nh})\right)h + \frac{1}{2}\left(g'(Y_n)g(Y_n) - g'(y_{nh})g(y_{nh})\right)\mathrm{I}_d h + \mathbb{E}_n\left[R_n\right]\right\|_2$$
$$\leq \|f'\|_\infty\|Y_n - y_{nh}\|_2\,h + \frac{1}{2}\left(\|g'\|_\infty^2 + \|g''\|_\infty\|g\|_\infty\right)\|Y_n - y_{nh}\|_2\,dh + \left\|\mathbb{E}_n\left[R_n\right]\right\|_2.$$

Similar to the proof of Theorem D.5, we use Young's inequality to estimate remainder terms.

$$\mathbb{E}\left[\left\|Y_n - y_{nh}\right\|_2\left\|\mathbb{E}_n\left[\left(Y_{n+1} - y_{(n+1)h}\right) - \left(Y_n - y_{nh}\right)\right]\right\|_2\right]$$
$$\leq \left\|Y_n - y_{nh}\right\|_{\mathbb{L}_2}^2\cdot O(h) + \mathbb{E}\left[\left\|Y_n - y_{nh}\right\|_2 h^{\frac{1}{2}}\left\|\mathbb{E}_n\left[R_n\right]\right\|_2 h^{-\frac{1}{2}}\right]$$
$$\leq \left\|Y_n - y_{nh}\right\|_{\mathbb{L}_2}^2\cdot O(h) + \frac{1}{2}\mathbb{E}\left[\left\|Y_n - y_{nh}\right\|_2^2 h + \left\|\mathbb{E}_n\left[R_n\right]\right\|_2^2 h^{-1}\right]$$
$$\leq \left\|Y_n - y_{nh}\right\|_{\mathbb{L}_2}^2\cdot O(h) + O(h^2),$$

where the final line is a consequence of Jensen's inequality as $\mathbb{E}\left[\|\mathbb{E}_n[R_n]\|_2^2\right] \leq \|R_n\|_{\mathbb{L}_2}^2 = O(h^3)$.

It is straightforward to estimate the second term in (25) using Minkowski's inequality as

$$\left\|\left(Y_{n+1} - y_{(n+1)h}\right) - \left(Y_n - y_{nh}\right)\right\|_{\mathbb{L}_2^n} = \left\|\left(f(Y_n) - f(y_{nh})\right)h + \left(g(Y_n) - g(y_{nh})\right)W_n\right.$$
$$+ \frac{1}{2}\left(g'(Y_n)g(Y_n) - g'(y_{nh})g(y_{nh})\right)W_n^{\otimes 2}$$
$$\left.- g'(y_{nh})g(y_{nh})\left(\mathbb{W}_n - \frac{1}{2}W_n^{\otimes 2}\right) + R_n\right\|_{\mathbb{L}_2^n}$$
$$\leq \|f'\|_\infty\left\|Y_n - y_{nh}\right\|_2 h + \|g'\|_\infty\left\|Y_n - y_{nh}\right\|_2\sqrt{d}\,h^{\frac{1}{2}}$$
$$+ \frac{1}{2}\left(\|g'\|_\infty^2 + \|g''\|_\infty\|g\|_\infty\right)\|Y_n - y_{nh}\|_2\|\mathcal{N}(0,\mathrm{I}_d)\|_{\mathbb{L}_4}^2 h$$
$$+ \|g'\|_\infty\|g\|_\infty\left\|\mathbb{W}_n - \frac{1}{2}W_n^{\otimes 2}\right\|_{\mathbb{L}_2^n} + \|R_n\|_{\mathbb{L}_2^n}$$
$$\leq \left\|Y_n - y_{nh}\right\|_2\cdot O(\sqrt{h}) + \left\|Y_n - Z_n\right\|_2\cdot O(\sqrt{h}) + O(h).$$

Therefore, by the $\mathbb{L}_4$-bound on $Y_n - Z_n$ given by Theorem D.6 and Young's inequality, we have
$$\left\|\left(Y_{n+1} - y_{(n+1)h}\right) - \left(Y_n - y_{nh}\right)\right\|_{\mathbb{L}_2}^2 \leq \left\|Y_n - y_{nh}\right\|_{\mathbb{L}_2}^2 \cdot O(h) + O\left(h^2\right).$$
Putting this all together, the inequality (25) becomes
$$\left\|Y_{n+1} - y_{(n+1)h}\right\|_{\mathbb{L}_2}^2 \leq \left(1 + O(h)\right)\left\|Y_n - y_{nh}\right\|_{\mathbb{L}_2}^2 + O\left(h^2\right),$$
and the result follows. $\qquad\square$

Just as before, we can immediately obtain a global error estimate by chaining together local estimates.

**Theorem D.12** (Global error estimate for the reversible Heun method). *Let $h_{\max} > 0$ be fixed. Then there exists a constant $C_2 > 0$ such that for all $n \in \{0, 1, \cdots, N\}$,*
$$\left\|Y_n - y_{nh}\right\|_{\mathbb{L}_2} \leq C_2\sqrt{h}\,,$$
*provided $h \leq h_{\max}$.*

*Proof.* Since $Y_0 = Z_0$, it follows from Theorem D.11 that
$$\left\|Y_n - y_{nh}\right\|_{\mathbb{L}_2}^2 \leq c_4 \sum_{k=0}^{n-1} e^{c_3 kh} h^2 = c_4 \frac{e^{c_3 nh} - 1}{e^{c_3 h} - 1} h^2 \leq c_4 \frac{e^{c_3 T} - 1}{e^{c_3 h} - 1} h^2 = O(h),$$
which gives the desired result. $\qquad\square$

## D.4   The reversible Heun method in the additive noise setting

We now replace the vector field $g$ in the Stratonovich SDE (13) with a fixed matrix $\sigma \in \mathbb{R}^{e \times d}$ to give
$$\mathrm{d}y_t = f(y_t)\,\mathrm{d}t + \sigma\,\mathrm{d}W_t\,. \tag{26}$$

Unsurprisingly, this simplifies the analysis and gives an $O(h)$ strong convergence rate for the method.

**Theorem D.13** (Taylor expansion of $Y$ when the SDE's noise is additive). *For $n \in \{0, 1\cdots, N-1\}$,*
$$Y_{n+1} = Y_n + f(Y_n)h + \sigma W_n + \frac{1}{2}f'(Y_n)(\sigma W_n)h + \widetilde{R}_n^Y,$$
*where the remainder term satisfies $\|\widetilde{R}_n^Y\|_{\mathbb{L}_2} = O(h^2)$.*

*Proof.* Expanding $f(Z_n)$ and $f(Z_{n+1})$ at $Y_n$ using Taylor's theorem [52, Theorem 3.5.6] yields
$$Y_{n+1} = Y_n + \frac{1}{2}\left(f(Z_n) + f(Z_{n+1})\right)h + \sigma W_n$$
$$= Y_n + f(Y_n)h + \sigma W_n + \frac{1}{2}f'(Y_n)(\sigma W_n)h + \widetilde{R}_n^Y\,,$$
where $\widetilde{R}_n^Y$ is given by
$$\widetilde{R}_n^Y := \frac{1}{2}f'(Y_n)f(Z_n)h^2 + \frac{1}{2}\int_0^1 (1-t)f''\left(Y_n + t\left(Z_n - Y_n\right)\right)\mathrm{d}t\left(Z_n - Y_n\right)^{\otimes 2}h$$
$$+ \frac{1}{2}\int_0^1 (1-t)f''\left(Y_n + t\left(Z_{n+1} - Y_n\right)\right)\mathrm{d}t\left(Z_{n+1} - Y_n\right)^{\otimes 2}h\,.$$
Similar to the proofs of Theorems D.3 and D.7, we can estimate this remainder term as
$$\|\widetilde{R}_n^Y\|_{\mathbb{L}_2}^2 \leq 2\left\|\frac{1}{2}f'(Y_n)f(Z_n)h^2\right\|_{\mathbb{L}_2}^2 + 2\left\|\widetilde{R}_n - \frac{1}{2}f'(Y_n)f(Z_n)h^2\right\|_{\mathbb{L}_2}^2$$
$$\leq \frac{1}{2}\|f'\|_\infty^2\|f\|_\infty^2 h^4 + \mathbb{E}\left[\left\|\int_0^1 (1-t)f''\left(Y_n + t\left(Z_n - Y_n\right)\right)\mathrm{d}t\left(Z_n - Y_n\right)^{\otimes 2}\right\|_{\mathbb{L}_2^n}^2\right]h^2$$
$$+ \mathbb{E}\left[\left\|\int_0^1 (1-t)f''\left(Y_n + t\left(Z_{n+1} - Y_n\right)\right)\mathrm{d}t\left(Z_{n+1} - Y_n\right)^{\otimes 2}\right\|_{\mathbb{L}_2^n}^2\right]h^2$$
$$\leq \frac{1}{2}\|f'\|_\infty^2\|f\|_\infty^2 h^4 + \mathbb{E}\left[\frac{1}{3}\|f''\|_\infty^2\left\|Z_n - Y_n\right\|_2^4\right]h^2 + \mathbb{E}\left[\frac{1}{3}\|f''\|_\infty^2\left\|Z_{n+1} - Y_n\right\|_{\mathbb{L}_4^n}^4\right]h^2$$
$$\leq \frac{1}{2}\|f'\|_\infty^2\|f\|_\infty^2 h^4 + 3\|f''\|_\infty^2\left\|Z_n - Y_n\right\|_{\mathbb{L}_2}^4 h^2 + \frac{8}{3}\|f''\|_\infty^2\mathbb{E}\left[\left\|f(Z_n)h + \sigma W_n\right\|_{\mathbb{L}_4^n}^4\right]h^2\,.$$

The result now follows by the $\mathbb{L}_4$-bound on $Y_n - Z_n$ given by Theorem D.6. $\qquad \square$

Likewise, the additive noise SDE (26) admits a simpler Taylor expansion than the general SDE (13).

**Theorem D.14** (Stochastic Taylor expansion for additive noise SDEs). *For $n \in \{0, 1, \cdots, N-1\}$,*

$$y_{(n+1)h} = y_{nh} + f(y_{nh})h + \sigma W_n + f'(y_{nh})\sigma J_n + \widetilde{R}_n^y,$$

*where $J_n$ denotes the time integral of Brownian motion over the interval $[nh, (n+1)h]$, that is*

$$J_n := \int_{nh}^{(n+1)h} \left(W_t - W_{nh}\right) \mathrm{d}t,$$

*and the remainder term satisfies $\|\widetilde{R}_n^y\|_{\mathbb{L}_2} = O(h^2)$.*

*Proof.* As $\sigma$ is constant, the terms involving second and third iterated integrals of $W$ do not appear. Therefore the result follows from more general expansions, such as [53, Proposition 5.10.1]. $\qquad \square$

To simplify the error analysis, we note the following lemma.

**Lemma D.15.** *For each $n \in \{0, 1, \cdots, N-1\}$, we define the random vector $H_n := \frac{1}{h}J_n - \frac{1}{2}W_n$. Then $H_n$ is independent of $W_n$ and $H_n \sim \mathcal{N}\left(0, \frac{1}{12}\mathrm{I}_d h\right)$.*

*Proof.* For the $d = 1$ case, the lemma was shown in [54, Definition 3.5]. When $d > 1$, the result is still straightforward as each coordinate of $W$ is an independent one-dimensional Brownian motion. $\qquad \square$

Using the same arguments as before, we can obtain error estimates for reversible Heun method.

**Theorem D.16** (Local error estimate for the reversible Heun method in the additive noise setting). *Let $h_{\max} > 0$ be fixed. Then there exist constants $c_5, c_6 > 0$ such that for $n \in \{0, 1, \cdots, N-1\}$,*

$$\left\|Y_{n+1} - y_{(n+1)h}\right\|_{\mathbb{L}_2}^2 \leq e^{c_5 h}\left\|Y_n - y_{nh}\right\|_{\mathbb{L}_2}^2 + c_6 h^3, \tag{27}$$

*provided $h \leq h_{\max}$.*

*Proof.* By Theorems D.13 and D.14 along with Lemma D.15, we have

$$
\begin{aligned}
Y_{n+1} - y_{(n+1)h} &= Y_n + f(Y_n)h + \sigma W_n + \frac{1}{2}f'(Y_n)(\sigma W_n)h + \widetilde{R}_n^Y \\
&\quad - y_{nh} - f(y_{nh})h - \sigma W_n - f'(y_{nh})\sigma\left(\frac{1}{2}W_n h + H_n h\right) - \widetilde{R}_n^y \\
&= Y_n - y_{nh} + \left(f(Y_n) - f(y_{nh})\right)h + \frac{1}{2}\left(f'(Y_n) - f'(y_{nh})\right)(\sigma W_n)h \\
&\quad - f'(y_{nh})(\sigma H_n)h + \widetilde{R}_n^Y - \widetilde{R}_n^y.
\end{aligned}
$$

The result now follows using exactly the same arguments as in the proof of Theorem D.11. $\qquad \square$

**Theorem D.17** (Global error estimate for the reversible Heun method in the additive noise setting). *Let $h_{\max} > 0$ be fixed. Then there exists a constant $C_3 > 0$ such that for all $n \in \{0, 1, \cdots, N\}$,*

$$\left\|Y_n - y_{nh}\right\|_{\mathbb{L}_2} \leq C_3\, h,$$

*provided $h \leq h_{\max}$.*

*Proof.* Since $Y_0 = Z_0$, it follows from Theorem D.16 that

$$\left\|Y_n - y_{nh}\right\|_{\mathbb{L}_2}^2 \leq c_6 \sum_{k=0}^{n-1} e^{c_5 kh} h^3 = c_6 \frac{e^{c_5 nh} - 1}{e^{c_5 h} - 1} h^3 \leq c_6 \frac{e^{c_5 T} - 1}{e^{c_5 h} - 1} h^3 = O(h^2),$$

which gives the desired result. $\qquad \square$

It is known that Heun's method achieves second order weak convergence for additive noise SDEs [55]. This can make Heun's method more appealing for SDE simulation than other two-stage methods, such as the standard midpoint method – which is first order weak convergent. Whilst understanding the weak convergence of the reversible Heun method is a topic for future work, we present numerical evidence that it has similar convergence properties as Heun's method for SDEs with additive noise.

We apply the standard and reversible Heun methods to the following scalar anharmonic oscillator:

$$\mathrm{d}y_t = \sin(y_t)\,\mathrm{d}t + \mathrm{d}W_t\,, \tag{28}$$

with $y_0 = 1$, and compute the following error estimates by standard Monte Carlo simulation:

$$S_N := \sqrt{\mathbb{E}\big[\big|Y_N - Y_T^{\text{fine}}\big|\big]},$$
$$E_N := \big|\mathbb{E}\big[Y_N\big] - \mathbb{E}\big[Y_T^{\text{fine}}\big]\big|,$$
$$V_N := \big|\mathbb{E}\big[Y_N^2\big] - \mathbb{E}\big[\big(Y_T^{\text{fine}}\big)^2\big]\big|,$$

where $\{Y_n\}$ denotes a numerical solution of the SDE (28) obtained with step size $h = \frac{T}{N}$ and $Y_T^{\text{fine}}$ is an approximation of $y_Y$ obtained by applying Heun's method to (28) with a finer step size of $\frac{1}{10}h$. Both $Y_N$ and $Y_T^{\text{fine}}$ are obtained using the same Brownian sample paths and the time horizon is $T = 1$.

The results of this simple numerical experiment are presented in Figures 5 and 6. From the graphs, we observe that the standard and reversible Heun methods exhibit very similar convergence rates (strong order 1.0 and weak order 2.0).

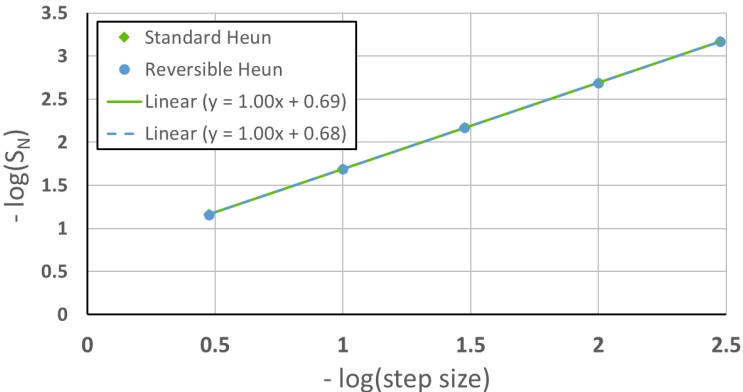

Figure 5: Log-log plot for the strong error estimator $S_N$ computed with $10^7$ Brownian sample paths.

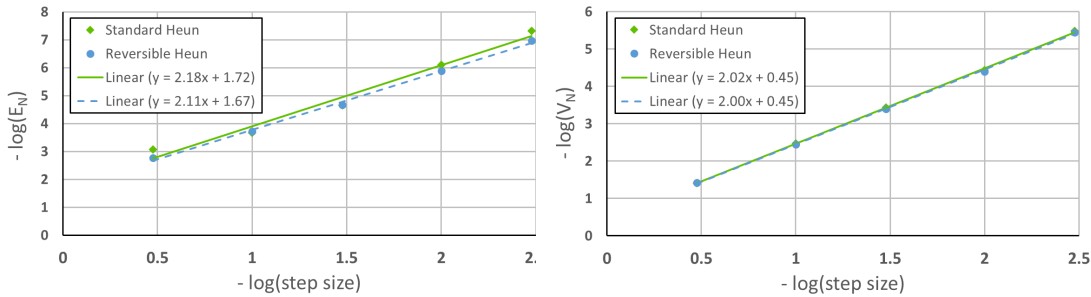

Figure 6: Log-log plots for the weak error estimators computed with $10^7$ Brownian sample paths.

### D.5   Stability properties of the reversible Heun method in the ODE setting

In this section, we present a stability result for the reversible Heun method when applied to an ODE,

$$y' = f(t, y). \tag{29}$$

Just as for the error analysis, it will be helpful to consider two steps of the reversible Heun method. In particular, the updates for the $Z$ component of the numerical solution satisfy

$$Z_{n+2} = Z_n + 2f(t_{n+1}, Z_{n+1})h, \tag{30}$$

with the second value of $Z$ being computed using a standard Euler step as $Z_1 := Z_0 + f(t_0, Z_0)h$. That is, $\{Z_n\}$ is precisely the numerical solution obtained by the leapfrog/midpoint method, see [56]. The absolute stability region of this ODE solver is well-known and given below.

**Theorem D.18** (Stability region of leapfrog/midpoint method [56, Section 2])**.** *Suppose that we apply the leapfrog/midpoint method to obtain a numerical solution $\{Z_n\}_{n\geq 0}$ for the linear test equation*

$$y' = \lambda y, \tag{31}$$

*where $\lambda \in \mathbb{C}$ with $\mathrm{Re}(\lambda) \leq 0$ and $y_0 \neq 0$. Then $|Z_n|$ is bounded for all $n$ if and only if $\lambda h \in [-i, i]$.*

Using similar techniques, it is straightforward to extend this result to the reversible Heun method.

**Theorem D.19** (Stability region of the reversible Heun method)**.** *Suppose that we apply the reversible Heun method to obtain a pair of numerical solutions $\{Y_n\}_{n\geq 0}, \{Z_n\}_{n\geq 0}$ for the linear test equation*

$$y' = \lambda y,$$

*where $\lambda \in \mathbb{C}$ with $\mathrm{Re}(\lambda) \leq 0$ and $y_0 \neq 0$. Then $\{Y_n, Z_n\}_{n\geq 0}$ is bounded if and only if $\lambda h \in [-i, i]$.*

*Proof.* By Theorem D.18, it is enough to show that $|Y_n|$ is bounded for all $n \geq 0$ when $\lambda h \in [-i, i]$. It follows from the difference equation (30) and the formula for $Z_1$ that

$$Z_n = \alpha \eta_1^n + \beta \eta_2^n,$$

where the constants $\alpha, \beta, \eta_1, \eta_2 \in \mathbb{C}$ are given by

$$\alpha := \frac{1}{2}y_0 \left(1 + \frac{1}{\sqrt{1 + \lambda^2 h^2}}\right),$$

$$\beta := \frac{1}{2}y_0 \left(1 - \frac{1}{\sqrt{1 + \lambda^2 h^2}}\right),$$

$$\eta_1 := \lambda h + \sqrt{1 + \lambda^2 h^2},$$

$$\eta_2 := \lambda h - \sqrt{1 + \lambda^2 h^2}.$$

For each $k \geq 0$, we have $Y_{k+1} = Y_k + \frac{1}{2}\lambda(Z_k + Z_{k+1})h$ and so we can explicitly compute $Y_n$ as

$$
\begin{aligned}
Y_n &= y_0 + \frac{1}{2}\lambda h \sum_{k=0}^{n-1}(Z_k + Z_{k+1}) \\
&= y_0 + \frac{1}{2}\lambda h \alpha(1 + \eta_1)\sum_{k=0}^{n-1}\eta_1^k + \frac{1}{2}\lambda h \beta(1 + \eta_2)\sum_{k=0}^{n-1}\eta_2^k \\
&= y_0 + \frac{1}{2}\lambda h \alpha\left(\frac{1 + \eta_1}{1 - \eta_1}\right)(1 - \eta_1^n) + \frac{1}{2}\lambda h \beta\left(\frac{1 + \eta_2}{1 - \eta_2}\right)(1 - \eta_2^n). \tag{32}
\end{aligned}
$$

Since $\lambda h \in [-i, i]$, we have $\eta_1 = \lambda h + \sqrt{1 - |\lambda h|^2}$ and $\eta_2 = \lambda h - \sqrt{1 - |\lambda h|^2}$, which implies that

$$|\eta_i|^2 = |\lambda h|^2 + (1 - |\lambda h|^2) = 1 \quad \text{and} \quad \mathrm{Im}(\eta_i) = |\lambda h|,$$

for both $i \in \{1, 2\}$. When $\lambda \neq 0$, it follows that $\eta_1, \eta_2 \in \{z \in \mathbb{C} : |z| = 1\} \setminus \{1\}$ and thus by (32), $|Y_n|$ is bounded for all $n \geq 0$. On the other hand, when $\lambda = 0$, we have $Y_n = y_0$ for all $n \geq 0$. $\quad\square$

**Remark D.20.** The reversible Heun method is not $A$-stable for ODEs as that would require $|\eta_i| < 1$.

**Remark D.21.** The domain $\{\lambda \in \mathbb{C} : \lambda h \in [-i, i]\}$ is also the absolute stability region for the (reversible) asynchronous leapfrog integrator proposed for Neural ODEs in Zhuang et al. [32].

# E    Sampling Brownian motion

## E.1    Algorithm

We begin by providing the complete traversal and splitting algorithm needed to find or create all intervals in the Brownian Interval, as in Section 4. See Algorithm 4.

Here, *List* is an ordered data structure that can be appended to, and iterated over sequentially. For example a linked list would suffice. We let `split_seed` denote a splittable PRNG as in Salmon et al. [34], Claessen and Pałka [35]. We use $*$ to denote an unfilled part of the data structure, equivalent to `None` in Python or a null pointer in C/C++; in particular this is used as a placeholder for the (nonexistent) children of leaf nodes. We use $=$ to denote the creation of a new local variable, and $\leftarrow$ to denote in-place modification of a variable.

## E.2    Discussion

The function `traverse` is a depth-first tree search for locating an interval within a binary tree. The search may split into multiple (potentially parallelisable) searches if the target interval crosses the intervals of multiple existing leaf nodes. If the search's target is not found then additional nodes are created as needed.

Sections 4 and E.1 now between them define the algorithm in technical detail.

There are some further technical considerations worth mentioning. Recall that the context we are explicitly considering is when sampling Brownian motion to solve an SDE forwards in time, then the adjoint backwards in time, and then discarding the Brownian motion. This motivates several of the choices here.

**Small intervals**    First, the access patterns of SDE solvers are quite specific. Queries will be over relatively small intervals: the step that the solver is making. This means that the list of nodes populated by `traverse` is typically small. In our experiments we observed it usually only consisting of a single element; occasionally two. In contrast if the Brownian Interval has built up a reasonable tree of previous queries, and was then queried over $[0, s]$ for $s \gg 0$, then a long (inefficient) list would be returned. It is the fact that SDE solvers do not make such queries that means this is acceptable.

**Search hints: starting from $\widehat{J}$**    Moreover, the queries are either just ahead (fixed-step solvers; accepted steps of adaptive-step solvers) or just before (rejected steps of adaptive-step solvers) previous queries. Thus in Algorithm 3, we keep track of the most recent node $\widehat{J}$, so that we begin `traverse` near to the correct location. This is what ensures the modal time complexity is only $\mathcal{O}(1)$, and not $\mathcal{O}(\log(1/s))$ in the average step size $s$, which for example would be the case if searching commenced from the root on every query.

**LRU cache**    The fact that queries are often close to one another is also what makes the strategy of using an LRU (least recently used) cache work. Most queries will correspond to a node that have a recently-computed parent in the cache.

**Backward pass**    The queries are broadly made left-to-right (on the forward pass), and then right-to-left (on the backward pass). (Other than the occasional rejected adaptive step.)

Left to its own devices, the forward pass will thus build up a highly imbalanced binary tree. At any one time, the LRU cache will contain only nodes whose intervals are a subset of some contiguous subinterval $[s, t]$ of the query space $[0, T]$. Letting $n$ be the number of queries on the forward pass, then this means that the backward pass will consume $\mathcal{O}(n^2)$ time – each time the backward pass moves past $s$, then queries will miss the LRU cache, and a full recomputation to the root will be triggered, costing $\mathcal{O}(n)$. This will then hold only nodes whose intervals are subsets of some contiguous subinterval $[u, s]$: once we move past $u$ then this $\mathcal{O}(n)$ procedure is repeated, $\mathcal{O}(n)$ times. This is clearly undesirable.

This is precisely analogous to the classical problem of optimal recomputation for performing back-propagation, whereby a dependency graph is constructed, certain values are checkpointed, and a minimal amount of recomputation is desired; see Griewank [57].

**Algorithm 4:** Definition of `traverse`

---

**def** `bisect`*(I : Node, x : $\mathbb{R}$)***:**
    # Only called on leaf nodes
    Let $I = ([a, b], s, I_{\texttt{parent}}, *, *)$
    $s_{\texttt{left}}, s_{\texttt{right}} = \texttt{split\_seed}(s)$
    $I_{\texttt{left}} = ([a, x], s_{\texttt{left}}, J, *, *)$
    $I_{\texttt{right}} = ([x, b], s_{\texttt{right}}, J, *, *)$
    $I \leftarrow ([a, b], s, I_{\texttt{parent}}, I_{\texttt{left}}, I_{\texttt{right}})$
    return

**def** `traverse_impl`*(I : Node, $[c, d]$ : Interval,* nodes *: List[Node])***:**
    Let $I = ([a, b], s, I_{\texttt{parent}}, I_{\texttt{left}}, I_{\texttt{right}})$

    # Outside our jurisdiction - pass to our parent
    **if** $c < a$ or $d > b$ **then**
        `traverse_impl`$(I_{\texttt{parent}}, [c, d], \text{nodes})$
        return
    # It's $I$ that is sought. Add $I$ to the list and return.
    **if** $c = a$ and $d = b$ **then**
        nodes.append($I$)
        return
    # Check if $I$ is a leaf or not.
    **if** $I_{left}$ is $*$ **then**
        # $I$ is a leaf
        **if** $a = c$ **then**
            # If the start points align then create children and add on the left child.
            # (Which is created in `bisect`.)
            `bisect`$(I, d)$
            nodes.append($I_{\texttt{left}}$)      # nodes is passed by reference
            return
        # Otherwise create children and pass on to our right child.
        # (Which is created in `bisect`.)
        `bisect`$(I, c)$
        `traverse_impl`$(I_{\texttt{right}}, [c, d], \text{nodes})$
        return
    **else**
        # $I$ is not a leaf.
        Let $I_{\texttt{left}} = ([a, m], s_{\texttt{left}}, I, I_{ll}, I_{lr})$
        **if** $d \leq m$ **then**
            # Strictly our left child's problem.
            `traverse_impl`$(I_{\texttt{left}}, [c, d], \text{nodes})$
            return
        **if** $c \geq m$ **then**
            # Strictly our right child's problem.
            `traverse_impl`$(I_{\texttt{right}}, [c, d], \text{nodes})$
            return
        # A problem for both of our children.
        `traverse_impl`$(I_{\texttt{left}}, [c, m], \text{nodes})$
        `traverse_impl`$(I_{\texttt{right}}, [m, d], \text{nodes})$
        return

**def** `traverse`*(I : Node, $[c, d]$ : Interval)***:**
    Let nodes be an empty List.
    `traverse_impl`$(I, [c, d], \text{nodes})$
    return nodes

In principle the same solution may be applied: apply a snapshotting procedure in which specific extra nodes are held in the cache. This is a perfectly acceptable solution, but implementing it requires some additional engineering effort, carefully determining which nodes to augment the cache with.

Fortunately, we have an advantage that Griewank [57] does not: we have some control over the dependency structure between the nodes, as we are free to prespecify any dependency structure we like. That is, we do not have to start the binary tree as just a stump. We may exploit this to produce an easier solution.

Given some estimate $\nu$ of the average step size of the SDE solver (which may be fixed and known if using a fixed step size solver), a size of the LRU cache $L$, and *before a user makes any queries*, then we simply make some queries of our own. These queries correspond to the intervals $[0, T/2], [T/2, T], [0, T/4], [T/4, T/2], \ldots$, so as to create a dyadic tree, such that the smallest intervals (the final ones in this sequence) are of size not more than $\nu L$. (In practice we use $\frac{4}{5}\nu L$ as an additional safety factor.)

Letting $[s, t]$ be some interval at the bottom of this dyadic tree, where $t \approx s + \frac{4}{5}\nu L$, then we are capable of holding every node within this interval in the LRU cache. Once we move past $s$ on the backward pass, then we may in turn hold the entire previous subinterval $[u, s]$ in the LRU cache, and in particular the values of the nodes whose intervals lie within $[u, s]$ may be computed in only logarithmic time, due to the dyadic tree structure.

This is now analogous to the Virtual Brownian Tree of Li et al. [15], Gaines and Lyons [58]. (Up to the use of intervals rather than points.) If desired, this approach may be loosely interpreted as placing a Brownian Interval on every leaf of a shallow Virtual Brownian Tree.

**Recursion errors**   We find that for some problems, the recursive computations of `traverse` (and in principle also `sample`, but this is less of an issue due to the LRU cache) can occasionally grow very deep. In particular this occurs when crossing the midpoint of the pre-specified tree: for this particular query, the traversal must ascend the tree to the root, and then descend all the way down again. As such `traverse` should be implemented with trampolining and/or tail recursion to avoid maximum depth recursion errors.

**CPU vs GPU memory**   We describe this algorithm as requiring only constant memory. To be more precise, the algorithm requires only constant GPU memory, corresponding to the fixed size of the LRU cache. As the Brownian Interval receives queries then its internal tree tracking dependencies will grow, and CPU memory will increase. For deep learning models, GPU memory is usually the limiting (and so more relevant) factor.

**Stochastic integrals**   What we have not discussed so far is the numerical simulation of integrals such as $\mathbb{W}_{s,t} = \int_s^t W_{s,r} \circ dW_r$ and $H_{s,t} = \frac{1}{t-s}\int_s^t \left(W_{s,r} - \left(\frac{r-s}{t-s}\right)W_{s,t}\right) dr$ which are used in higher order SDEs solvers (for example, the Runge-Kutta methods in [59] and the log-ODE method in [54]). Just like increments $W_{s,t}$, these integrals fit nicely into an interval-based data structure.

In general, simulating the pair $(W_{s,t}, \mathbb{W}_{s,t})$ is known to be a difficult problem [60], and exact algorithms are only known when $W$ is one or two dimensional [61]. However, the approximation proposed in [62] and further developed in [63, 64] constitutes a simple and computable solution. Their approach is to generate

$$\widetilde{\mathbb{W}}_{s,t} := \frac{1}{2}W_{s,t} \otimes W_{s,t} + H_{s,t} \otimes W_{s,t} - W_{s,t} \otimes H_{s,t} + \lambda_{s,t},$$

where $\lambda_{s,t}$ is an anti-symmetric matrix with independent entries $\lambda_{s,t}^{i,j} \sim \mathcal{N}(0, \frac{1}{12}(t-s)^2), i < j$.

In these works, the authors input the pairs $\{(W_{t_n,t_{n+1}}, \widetilde{\mathbb{W}}_{t_n,t_{n+1}})\}_{0 \leq n \leq N-1}$ into a SDE solver (the Milstein and log-ODE methods respectively) and prove that the resulting approximation achieves a 2-Wasserstein convergence rate close to $O(1/N)$, where $N$ is the number of steps. In particular, this approach is efficient and avoids the use of costly Lévy area approximations, such as in [65, 66, 67].

# F   Experimental Details and Further Results

## F.1   Metrics

Several metrics were used to evaluate final model performance of the trained Latent SDEs and SDE-GANs.

**Real/fake classification**   A classifier was trained to distinguish real from generated data. This is trained by taking an 80%/20% split of the test data, training the classifier on the 80%, and evaluating its performance on the 20%. This produces a classification accuracy as a performance metric.

We parameterise the classifier as a Neural CDE [45], whose vector field is an MLP with two hidden layers each of width 32. The evolving hidden state is also of width 32. A final classification result is given by applying a learnt linear readout to the final hidden state, which produces a scalar. A sigmoid is then applied and this is trained with binary cross-entropy.

It is trained for 5000 steps using Adam with a learning rate of $10^{-4}$ and a batch size of 1024.

*Smaller accuracies – indicating inability to distinguish real from generated data – are better.*

**Label classification (train-on-synthetic-test-on-real)**   Some datasets (in particular the air quality dataset) have labelled classes associated with each sample time series. For these datasets, a classifier was trained on the generated data – possible as every model we train is trained conditional on the class label as an input – and then evaluated on the real test data. This produces a classification accuracy as a performance metric.

We parameterise the classifier as a Neural CDE, with the same architecture as before. A final classification result is given by applying a learnt linear readout to the final hidden state, which produces a vector of unnormalised class probabilities. These are normalised with a softmax and trained using cross-entropy.

It is trained for 5000 steps using Adam with a learning rate of $10^{-4}$ and a batch size of 1024.

*Larger accuracies – indicating similarity of real and generated data – are better.*

**Prediction (train-on-synthetic-test-on-real)**   A sequence-to-sequence model is trained to perform time series forecasting: given the first 80% of a time series, can the latter 20% be predicted. This is trained on the generated data, and then evaluated on the real test data. This produces a regression loss as a performance metric.

We parameterise the predictor as a sequence-to-sequence Neural CDE / Neural ODE pair. The Neural CDE is as before. The Neural ODE has a vector field which is an MLP of two hidden layers, each of width 32. Its evolving hidden state is also of width 32. An evolving prediction is given by applying a learnt linear readout to the evolving hidden state, which produces a time series of predictions. These are trained using an $L^2$ loss.

It is trained for 5000 steps using Adam with a learning rate of $10^{-4}$ and a batch size of 1024.

*Smaller losses – indicating similarity of real and generated data – are better.*

**Maximum mean discrepancy**   Maximum mean discrepancies [68] can be used to compute a (semi)distance between probability distributions. Given some set $\mathcal{X}$, a fixed feature map $\psi \colon \mathcal{X} \to \mathbb{R}^m$, a norm $\| \cdot \|$ on $\mathbb{R}^m$, and two probability distributions $\mathbb{P}$ and $\mathbb{Q}$ on $\mathcal{X}$, this is defined as

$$\| \mathbb{E}_{P \sim \mathbb{P}} [\psi(P)] - \mathbb{E}_{Q \sim \mathbb{Q}} [\psi(Q)] \| .$$

In practice $\mathbb{P}$ corresponds to the true distribution, of which we observe samples of data, and $\mathbb{Q}$ corresponds to the law of the generator, from which we may sample arbitrarily many times. Given $N$ empirical samples $P_i$ from the true distribution, and $M$ generated samples $Q_i$ from the generator, we may thus approximate the MMD distance via

$$\left\| \frac{1}{N} \sum_{i=1}^{N} \psi(P_i) - \frac{1}{M} \sum_{i=1}^{M} \psi(Q_i) \right\| .$$

In our case, $\mathcal{X}$ corresponds to the observed time series, and we use a depth-5 signature transform as the feature map [69, 70]. Similarly, the untruncated signature can be used as a feature map [71, 72, 73, 74, 75].

(Note that MMDs may also be used as differentiable optimisation metrics provided $\psi$ is differentiable [76, 77]. A mistake we have seen 'in the wild' for training SDEs is to choose a feature map that is overly simplistic, such as taking $\psi$ to be the marginal mean and variance at all times. Such a feature map would fail to capture time-varying correlations; for example $W$ and $t \mapsto W(0)\sqrt{t}$, where $W$ is a Brownian motion, would be equivalent under this feature map.)

*Smaller values – indicating similarity of real and generated data – are better.*

## F.2   Common details

The following details are in common to all experiments.

**Libraries used**   PyTorch was used as an autodifferentiable tensor framework [27].

SDEs were solved using `torchsde` [42]. CDEs were solved using `torchcde` [78]. ODEs were solved using `torchdiffeq` [79]. Signatures were computed using Signatory [74].

Tensors had their shapes annotated using the torchtyping [49] library, which helped to enforce correctness of the implementation.

Hyperparameter optimisation was performed using the Ax library [80].

**Numerical methods**   SDEs were solved using either the reversible Heun method or the midpoint method (as per the experiment), and trained using continuous adjoint methods.

The CDE used in the discriminator of an SDE-GAN was solved using either the reversible Heun method, or the midpoint method, in common with the choice made in the generator, and trained using continuous adjoint methods.

The ODEs solved for the train-on-synthetic-test-on-real prediction metric used the midpoint method, and were trained using discretise-then-optimise backpropagation. The CDEs solved for the various evaluation metrics used the midpoint method, and trained using discretise-then-optimise backpropagation. (These were essentially arbitrary choices – we merely needed to fix some choices throughout to ensure a fair comparison.)

**Normalisation**   Every dataset is normalised so that its initial value (at time $= 0$) has mean zero and unit variance. (That is, calculate mean and variance statistics of just the initial values in the dataset, and then normalise every element of the dataset using these statistics.)

We speculate that normalising based on the initial condition produces better results than calculating mean and variance statistics over the whole trajectory, as the rest of the trajectory cannot easily be learnt unless its initial condition is well learnt first. We did not perform a thorough investigation of this topic, merely finding that this worked well enough on the problems considered here.

The times at which observations were made were normalised to have mean zero and unit range. (This is of relevance to the modelling, as the generated samples must be made over the same timespan: that is the integration variable $t$ must correspond to some parameterisation of the time at which data is actually observed.)

**Dataset splits**   We used 70% of the data for training, 15% for validation and hyperparameter optimisation, and 15% for testing.

**Optimiser**   The batch size is always taken to be 1024. The number of training steps varies by experiment, see below.

We use Adam [81] to train every Latent SDE.

Following Kidger et al. [16] we use Adadelta [82] to train every SDE-GAN.

**Stochastic weight averaging**  When training SDE-GANs, we take a Cesàro mean over the latter 50% of the training steps, of the generator's weights, to produce the final trained model. Known as 'stochastic weight averaging' this often slightly improves GAN training [83, 84].

**Architectures**  Each of $\zeta_\theta, \mu_\theta, \sigma_\theta, \xi_\phi, f_\phi, g_\phi$ were parameterised as MLPs. (From equations (1), (2), (10).)

Following Li et al. [15], then $\nu_\phi$ (of equation (10)) was parameterised as an MLP composed with a GRU.

In brief, that is $\nu_\phi(t, \widehat{X}_t, Y_{\text{true}}) = \nu_\phi^1(t, \widehat{X}_t, \nu_\phi^2(Y_{\text{true}}|_{[t,T]}))$, where $\nu_\phi^1$ is an MLP, and $\nu_\phi^2$ is a GRU run backwards-in-time from $T$ to $t$ over whatever discretisation of $Y_{\text{true}}|_{[t,T]}$ is observed.

In all cases, for simplicity, the LipSwish activation function was used throughout.

**Hyperparameter optimisation**  Hyperparameter optimisation used the default optimisation strategy provided by Ax (initial quasirandom Sobol sampling followed by Bayesian optimisation). The optimisation metric was the MMD evaluation metric, due to the speed at which it can be computed relative to the other optimisation metrics (which require training an auxiliary model).

The Latent SDE model was hyperoptimised on every dataset. To ensure we do not bias results in our favour, the hyperoptimised models used the midpoint method, not the reversible Heun method, throughout, and was optimised using discretise-then-optimise backpropagation.

The SDE-GAN models then used the same hyperparameters where applicable, for example on learning rate, neural network size, and so on. This fixes mosts of the hyperparameters. A few extra hyperparameters were then chosen manually.

First, the size of the initial noise $V$ and the dimensionality of the Brownian motion $W$ were arbitrarily fixed at 10.

Second, we adjusted the initialisation strategy for the parameters of the SDE. For each dataset, we picked some constants

$$\alpha > 0, \quad \beta > 0, \tag{33}$$

and multiplied the parameters $\theta$ at initialisation by either $\alpha$ or $\beta$, depending on whether the parameters were used to generate the initial condition ($\zeta_\theta$), or were used in the vector fields of the SDE ($\mu_\theta, \sigma_\theta, \ell_\theta$), respectively. This helped to ensure that the SDE had a good initialisation, and therefore took fewer steps to train. The values $\alpha, \beta$ were chosen by manually plotting generated samples from an untrained model against real data. (A less naïve approach would be nice.)

The one exception is the Ornstein–Uhlenbeck dataset, for which the SDE-GAN had $\zeta_\theta$, $\mu_\theta$, $\sigma_\theta, \xi_\phi, f_\phi, g_\phi$ parameterised as MLPs with one hidden layer of width 32, and had an evolving hidden state $X$ of size 32; these figures were chosen as being known to work based on early experiments.

**Compute resources**  Experiments were performed on an internal GPU cluster. Each experiment used only a single GPU at a time. Amount of compute time varied depending on the experiment – some took a few hours, some took a few days. Precise times given in the tables of results.

(Moreover some would likely have taken a few weeks without the algorithmic speed improvements introduced in this paper. Recall that each baseline experiment used the improvements introduced in the other sections of this paper, simply to produce tractable training times. For example the Brownian Interval was used throughout.)

GPU types varied between GeForce RTX 2080 Ti, Quadro GP100, and A100s.

### F.3  Weights dataset

**Technical details**  We consider a dataset of weights of a small convolutional network, as it is trained to classify MNIST, with training using stochastic gradient descent, as in Kidger et al. [16]. The network was trained 10 times, and all weight trajectories, across all runs, were aggregated together to form a dataset of univariate time series.

Each time series is 50 elements long, corresponding to how the weights change over 50 epochs.

Table 4: SDE-GAN on weights dataset. Mean $\pm$ standard deviation averaged over three runs.

| Solver | Test Metrics | | | |
| --- | --- | --- | --- | --- |
| | Real/fake classification accuracy (%) | Prediction loss | MMD ($\times 10^{-2}$) | Training time (days) |
| Midpoint | $77.0 \pm 6.9$ | $0.303 \pm 0.369$ | $4.38 \pm 0.67$ | $5.12 \pm 0.01$ |
| Reversible Heun | $\mathbf{75.5 \pm 0.01}$ | $\mathbf{0.068 \pm 0.037}$ | $\mathbf{1.75 \pm 0.3}$ | $\mathbf{2.59 \pm 0.05}$ |

We train an SDE-GAN on this dataset. We train the generator and discriminator for $80\,000$ steps each.

Each MLP ($\zeta_\theta, \mu_\theta, \sigma_\theta, \xi_\phi, f_\phi, g_\phi$) is parameterised as having two hidden layers each of width 67. The evolving hidden states $X, H$ are each of size 62.

The parameters of $\zeta_\theta, \xi_\phi$ have a learning rate of $4.9 \times 10^{-3}$. The parameters of $\mu_\theta, \sigma_\theta, f_\phi, g_\phi$ have a learning rate of $1.3 \times 10^{-3}$.

The initialisation scaling parameters $\alpha$ and $\beta$ (of equation (33)) were selected to be 4.5 and 0.25 respectively.

**Results**   We compare the reversible Heun method to the midpoint method on the weights dataset, by training an SDE-GAN.

We compute three test metrics: classification of real versus generated data, forecasting via train-on-synthetic-test-on-real, and a maximum mean discrepancy. We additionally report training time. See Table 4.

First and most notably, we see a dramatic reduction in training time: the training time of the reversible Heun method is roughly half that of the midpoint method. This corresponds to the reduction in vector field evaluations of the reversible Heun method.

We additionally see better performance on the test metrics, as compared to the midpoint method. This corresponds to the calculation of numerically precise gradients via the reversible Heun method.

We believe that these test metrics could be further improved (in particular the classification accuracy) given further training time.

The Brownian Interval (Section 4) is used to sample Brownian noise, and the SDE-GAN is trained using clipping as in Section 5. (Without either of which the baseline experiments would have taken infeasibly long to run.)

### F.4   Air quality dataset

**Technical details**   This is a dataset of air quality samples over Beijing, as they vary over the course of a day. Each time series is 24 elements long, corresponding to a single hour each day. We consider specifically the PM2.5 particulate matter concentration, and ozone concentration, to produce a dataset of bivariate time series. In particular the ozone channel was selected as displaying obvious non-autonomous behaviour: the latter half of the time series often includes a peak.

This dataset is available via the UCI machine learning repository [85, 86].

Each time series has a label, corresponding to which of 12 different locations the measurements were made at.

We train a Latent SDE on this dataset. We train for $40\,000$ steps.

Each MLP ($\zeta_\theta, \mu_\theta, \sigma_\theta, \xi_\phi, \nu_\phi^1$) is parameterised as having a single hidden layer of width 84. The evolving hidden state $X$ is of size 63.

$\nu_\phi^2$ was parameterised as GRU with hidden size 84, whose final hidden state has a learnt affine map applied, to produce vector of size 60.

Table 5: Latent SDE on air quality dataset. Mean ± standard deviation averaged over three runs.

| Solver | Test Metrics | | | | |
| --- | --- | --- | --- | --- | --- |
| | Real/fake classification accuracy (%) | Label classification accuracy (%) | Prediction loss | MMD $(\times 10^{-3})$ | Training time (hours) |
| Midpoint | **92.3 ± 0.02** | 46.3 ± 5.1 | **0.281 ± 0.009** | 5.91 ± 2.06 | 5.58 ± 0.54 |
| Reversible Heun | 96.7 ± 0.01 | **49.2 ± 0.02** | 0.314 ± 0.005 | **4.72 ± 2.90** | **4.47 ± 0.31** |

The parameters of $\zeta_\theta$ have learning rate $1.1 \times 10^{-4}$. The parameters of $\mu_\theta, \sigma_\theta, \xi_\phi, \nu_\phi^1, \nu_\phi^2$ have learning rate $1.9 \times 10^{-5}$.

The initialisation scaling parameters $\alpha$ and $\beta$ (of equation (33)) were selected to be 2 and 1 respectively.

**Results**    We compare the reversible Heun method to the midpoint method on the air quality dataset, by training a Latent SDE.

We compute four test metrics: classification of real versus generated data, classification via train-on-synthetic-test-on-real, forecasting via train-on-synthetic-test-on-real, and a maximum mean discrepancy. We additionally report training time. See Table 5.

Here, the most important metric is again the improvement in training time: whilst less dramatic than the previous SDE-GAN experiment, it is still a speed improvement of $1.2\times$.

Performance on the test metrics varies between the solvers, without a clear pattern.

It is worth noting the apparently poor real/fake classification accuracies obtained using either solver. This is typical of Latent SDEs in general, in particular as opposed to SDE-GANs. Whilst Latent SDEs are substantially quicker to train, they tend to produce less convincing samples.

### F.5   Gradient error analysis

We investigate the error made in the gradient calculation using continuous adjoints. We consider using the midpoint method, Heun's method, and the reversible Heun method, and vary the step size in decreasing powers of two.

Our test problem is to compute $dX_1/dX_0$ and $dX_1/d\theta$ over a batch of 32 samples of

$$dX_t = f_\theta(t, X_t)\, dt + g_\theta(t, X_t) \circ dW_t,$$

where $X_t, f_\theta(t, X_t) \in \mathbb{R}^{32}, W_t \in \mathbb{R}^{16}, g_\theta(t, X_t) \in \mathbb{R}^{32,16}$ and $f_\theta$, $g_\theta$ are feedforward neural networks with a single hidden layer of width 8, and LipSwish activation function. Additionally $f_\theta$ has a $\tanh$ as a final nonlinearity, and $g_\theta$ has a sigmoid as final nonlinearity.

We compare the gradients computed via optimise-then-discretise against discretise-then-optimise, and plot the relative $L^1$ error. Letting $\delta_{o-d} \in \mathbb{R}^M$ and $\delta_{d-o} \in \mathbb{R}^M$ denote these gradients (with $M$ equal to the size of $X_0$ plus the size of $\theta$), then this is defined as

$$\frac{\sum_{i=1}^{M} \left| \delta_{o-d}^i - \delta_{d-o}^i \right|}{\max\{\sum_{i=1}^{M} \left| \delta_{o-d}^i \right|, \sum_{i=1}^{M} \left| \delta_{d-o}^i \right|\}}.$$

Numerical values are shown in Table 6. Results are plotted graphically in the main text.

### F.6   Brownian benchmarks

We benchmark the Brownian Interval against the Virtual Brownian Tree across several benchmarks.

**Access benchmarks**    We subdivide the interval $[0, 1]$ into a disjoint union of equal-sized intervals. Across each interval we then place a query for a small Brownian increment.

We consider subdividing $[0, 1]$ into different numbers of subintervals (and so of different sizes): either 10, 100 or 1000 subintervals.

We consider several access patterns.

Sequential access: this involves querying every interval in order, from 0 to 1. Every interval is queried precisely once. This simulates an SDE solve from 0 to 1.

Doubly sequential access: this involves querying every interval in order, from 0 to 1, and then querying them again in reverse order, from 1 to 0. Every interval is queried precisely twice. This simulates an SDE solve from 0 to 1, followed by a backpropagation via the continuous adjoint method, from 1 to 0.

Random access: this involves querying every interval precisely once, in a random order.

We consider several batch sizes: either a single Brownian simulation, a typical batch size of 2560 simultaneous Brownian simulations (corresponding to a batch of size 256, each with a vector of 10 Brownian motions), and a large batch size of 32768 simultaneous Brownian simulations.

For every such combination we report metrics for the Brownian Interval and the Virtual Brownian Tree.

For every such combination we run 32 repeats. The reported metric is the fastest (minimum time) over these repeats.[8]

See Tables 7, 8, and 9.

The Brownian Interval is consistently and substantially faster than the Virtual Brownian Tree, across all access patterns, batch sizes, and number of subintervals.

On the doubly sequential access benchmark (emulating the SDE solve and backpropagation typical in practice), we see that total speed-ups vary from a factor of $2.89\times$ to a factor of $13.5\times$. That is, at minimum, speed is roughly tripled. Potentially it is improved by over an order of magnitude. Typical values are speed-ups of $6$–$8\times$.

**SDE solve benchmarks**   We now benchmark the Brownian Interval against the Virtual Brownian Tree on the actual task of solving and backpropagating through an SDE.

As before we consider several numbers of subintervals, several batch sizes, and run 32 repeats and take the fastest time.

Our test SDE is an Itô SDE with diagonal noise:

$$\mathrm{d}X_t^i = \tanh(A^{i,j} X_t^j)\,\mathrm{d}t + \delta^{i,k}\delta^{i,l}\tanh(B^{k,j} X_t^j)\,\mathrm{d}W_t^l,$$

where either $X_t, W_t \in \mathbb{R}^1$, $X_t, W_t \in \mathbb{R}^{10}$, or $X_t, W_t \in \mathbb{R}^{16}$, corresponding to the small, medium and large batch sizes considered in the previous set of benchmarks. Likewise $A, B \in \mathbb{R}^{1\times 1}$, $A, B \in \mathbb{R}^{10\times 10}$, or $A, B \in \mathbb{R}^{16\times 16}$ are random matrices. $\tanh$ is applied component-wise.

For $i = 10, 100, 1000$ subintervals, we calculate a forward pass for $[0, 1]$ using the Euler–Maruyama method, to calculate $X_{j/(i-1)}$ for $j \in \{0, \ldots, i-1\}$. We then backpropagate from the vector $(X_{j/(i-1)})_j$ to $X_0$, using the continuous adjoint method.

See Table 10.

We once again see that the Brownian Interval is uniformly and substantially faster than the Virtual Brownian Tree, in all regimes. On smaller problems (batch size equal to 1 or 2560), then it is typically twice as fast; at worst it is $1.89\times$ as fast. Meanwhile on larger problems (batch size equal to 32768), then it is typically ten times as fast.

These benchmarks include the realistic overheads involved in solving an SDE (such as evaluating its vector fields), and represent typical speed-ups from using the Brownian Interval over the Virtual Brownian Tree.

---

[8]Not the mean. Errors in speed benchmarks are one-sided, and so the minimum time represents the least noisy measurement.

Table 6: Relative $L^1$ error on the gradient analysis test problem.

| Method | Step size | | | |
| --- | --- | --- | --- | --- |
| | $2^0$ | $2^{-2}$ | $2^{-4}$ | $2^{-6}$ |
| Midpoint | $9.4 \times 10^{-2}$ | $1.1 \times 10^{-2}$ | $2.6 \times 10^{-3}$ | $7.2 \times 10^{-4}$ |
| Heun | $1.4 \times 10^{-1}$ | $5.9 \times 10^{-2}$ | $1.4 \times 10^{-2}$ | $2.9 \times 10^{-3}$ |
| Reversible Heun | $\mathbf{1.9 \times 10^{-17}}$ | $\mathbf{2.7 \times 10^{-16}}$ | $\mathbf{3.3 \times 10^{-16}}$ | $\mathbf{5.7 \times 10^{-16}}$ |

| $2^{-8}$ | $2^{-10}$ | $2^{-14}$ |
| --- | --- | --- |
| $1.8 \times 10^{-4}$ | $4.8 \times 10^{-5}$ | $2.5 \times 10^{-6}$ |
| $8.2 \times 10^{-4}$ | $2.8 \times 10^{-4}$ | $1.3 \times 10^{-5}$ |
| $\mathbf{1.0 \times 10^{-15}}$ | $\mathbf{1.8 \times 10^{-15}}$ | $\mathbf{6.5 \times 10^{-15}}$ |

Table 7: Speed on the sequential access benchmark. Minimum over 32 runs.

| Batch size, subinterval number | Speed (seconds) | |
| --- | --- | --- |
| | Virtual Brownian Tree | Brownian Interval |
| 1, 10 | $8.08 \times 10^{-3}$ | $\mathbf{2.59 \times 10^{-3}}$ |
| 1, 100 | $7.84 \times 10^{-2}$ | $\mathbf{3.12 \times 10^{-2}}$ |
| 1, 1000 | $7.96 \times 10^{-1}$ | $\mathbf{3.27 \times 10^{-1}}$ |
| 2560, 10 | $1.90 \times 10^{-2}$ | $\mathbf{4.13 \times 10^{-3}}$ |
| 2560, 100 | $1.94 \times 10^{-1}$ | $\mathbf{4.95 \times 10^{-2}}$ |
| 2560, 1000 | $1.93 \times 10^{0}$ | $\mathbf{5.15 \times 10^{-1}}$ |
| 32768, 10 | $1.16 \times 10^{-1}$ | $\mathbf{1.71 \times 10^{-2}}$ |
| 32768, 100 | $1.40 \times 10^{0}$ | $\mathbf{2.02 \times 10^{-1}}$ |
| 32768, 1000 | $1.43 \times 10^{1}$ | $\mathbf{2.13 \times 10^{0}}$ |

Table 8: Speed on the doubly sequential access benchmark. Minimum over 32 runs.

| Batch size, subinterval number | Speed (seconds) | |
| --- | --- | --- |
| | Virtual Brownian Tree | Brownian Interval |
| 1, 10 | $2.02 \times 10^{-2}$ | $\mathbf{2.79 \times 10^{-3}}$ |
| 1, 100 | $2.42 \times 10^{-1}$ | $\mathbf{4.96 \times 10^{-2}}$ |
| 1, 1000 | $1.67 \times 10^{0}$ | $\mathbf{5.79 \times 10^{-1}}$ |
| 2560, 10 | $3.23 \times 10^{-2}$ | $\mathbf{4.31 \times 10^{-3}}$ |
| 2560, 100 | $3.91 \times 10^{-1}$ | $\mathbf{8.05 \times 10^{-2}}$ |
| 2560, 1000 | $4.01 \times 10^{0}$ | $\mathbf{9.56 \times 10^{-1}}$ |
| 32768, 10 | $2.30 \times 10^{-1}$ | $\mathbf{1.70 \times 10^{-2}}$ |
| 32768, 100 | $2.92 \times 10^{0}$ | $\mathbf{3.49 \times 10^{-1}}$ |
| 32768, 1000 | $2.91 \times 10^{1}$ | $\mathbf{4.36 \times 10^{0}}$ |

Table 9: Speed on the random access benchmark. Minimum over 32 runs.

| Batch size, subinterval number | Speed (seconds) | |
| --- | --- | --- |
| | Virtual Brownian Tree | Brownian Interval |
| 1, 10 | $1.53 \times 10^{-2}$ | $\mathbf{2.56 \times 10^{-3}}$ |
| 1, 100 | $1.09 \times 10^{-1}$ | $\mathbf{5.17 \times 10^{-2}}$ |
| 1, 1000 | $8.52 \times 10^{-1}$ | $\mathbf{1.02 \times 10^{0}}$ |
| 2560, 10 | $1.50 \times 10^{-2}$ | $\mathbf{3.56 \times 10^{-3}}$ |
| 2560, 100 | $1.86 \times 10^{-1}$ | $\mathbf{8.96 \times 10^{-2}}$ |
| 2560, 1000 | $1.99 \times 10^{0}$ | $\mathbf{1.80 \times 10^{0}}$ |
| 32768, 10 | $1.18 \times 10^{-1}$ | $\mathbf{1.60 \times 10^{-2}}$ |
| 32768, 100 | $1.47 \times 10^{0}$ | $\mathbf{3.63 \times 10^{-1}}$ |
| 32768, 1000 | $1.44 \times 10^{1}$ | $\mathbf{9.08 \times 10^{0}}$ |

Table 10: Speed on the sequential access benchmark. Minimum over 32 runs.

| Batch size, subinterval number | Speed (seconds) | |
| --- | --- | --- |
| | Virtual Brownian Tree | Brownian Interval |
| 1, 10 | $1.42 \times 10^{-1}$ | $\mathbf{7.18 \times 10^{-2}}$ |
| 1, 100 | $1.55 \times 10^{0}$ | $\mathbf{8.16 \times 10^{-1}}$ |
| 1, 1000 | $1.61 \times 10^{1}$ | $\mathbf{8.52 \times 10^{0}}$ |
| 2560, 10 | $2.61 \times 10^{-1}$ | $\mathbf{1.13 \times 10^{-1}}$ |
| 2560, 100 | $3.00 \times 10^{0}$ | $\mathbf{1.31 \times 10^{0}}$ |
| 2560, 1000 | $3.00 \times 10^{1}$ | $\mathbf{1.31 \times 10^{1}}$ |
| 32768, 10 | $4.34 \times 10^{1}$ | $\mathbf{4.66 \times 10^{0}}$ |
| 32768, 100 | $4.99 \times 10^{2}$ | $\mathbf{4.68 \times 10^{1}}$ |
| 32768, 1000 | $1.54 \times 10^{3}$ | $\mathbf{4.74 \times 10^{2}}$ |

### F.7 Time-dependent Ornstein–Uhlenbeck dataset

**Technical details**   This is a dataset of univariate samples of length 32 from the time-dependent Ornstein–Uhlenbeck process

$$\mathrm{d}Y_{\mathrm{true},t} = (\rho t - \kappa Y_{\mathrm{true},t})\,\mathrm{d}t + \chi\,\mathrm{d}W_t,$$

with $\rho = 0.02$, $\kappa = 0.1$ and $\chi = 0.4$ and $t \in [0, 31]$

We train an SDE-GAN on this dataset. We train the generator for 20 000 steps and the discriminator for 100 000 steps.

Each MLP ($\zeta_\theta, \mu_\theta, \sigma_\theta, \xi_\phi, f_\phi, g_\phi$) is parameterised as having a single hidden layer of width 32. The evolving hidden states $X$, $H$ are each of size 32.

The parameters of $\zeta_\theta, \xi_\phi$ have a learning rate of $1.6 \times 10^{-3}$. The parameters of $\mu_\theta, \sigma_\theta, f_\phi, g_\phi$ have a learning rate of $2.0 \times 10^{-4}$.

The initialisation scaling parameters $\alpha$ and $\beta$ (of equation (33)) were selected to be 5 and 0.5 respectively.

**Results**   We compare training SDE-GANs by careful clipping (as in Section 5) to using gradient penalty (as in Kidger et al. [16]) on the OU dataset.

As our implementation of the reversible Heun method does not support a double backward, we provide two comparisons of interest: reversible Heun method with clipping against midpoint with gradient penalty, and midpoint with clipping against midpoint with gradient penalty.

Table 11: SDE-GAN on OU dataset. Mean $\pm$ standard deviation averaged over three runs.

| Solver | Test Metrics | | | |
| --- | --- | --- | --- | --- |
| | Real/fake classification accuracy (%) | Prediction loss | MMD ($\times 10^{-1}$) | Training time (hours) |
| Midpoint with gradient penalty | $98.2 \pm 2.4$ | $2.71 \pm 1.03$ | $2.58 \pm 1.81$ | $55.0 \pm 27.7$ |
| Midpoint with clipping | $93.9 \pm 6.9$ | $1.65 \pm 0.17$ | $1.03 \pm 0.10$ | $32.5 \pm 12.1$ |
| Reversible Heun with clipping | $\mathbf{67.7 \pm 1.1}$ | $\mathbf{1.38 \pm 0.06}$ | $\mathbf{0.45 \pm 0.22}$ | $\mathbf{29.4 \pm 8.9}$ |

We compute three test metrics: classification of real versus generated data, forecasting via train-on-synthetic-test-on-real, and a maximum mean discrepancy. We additionally report training time. See Table 11.

We see that reversible Heun with clipping dominates midpoint with clipping, which in turn dominates midpoint with gradient penalty, across all metrics.

As per Kidger et al. [16], the poor performance of gradient penalty is due in part to the numerical errors of a double adjoint. Switching to midpoint with clipping produces substantially better test metrics. It additionally improves training speed by $1.41\times$.

Switching from midpoint with clipping to reversible Heun with clipping then produces another substantial boost to the test metrics – most notably the real-versus-fake classification accuracy. It also improves training speed by another $1.09\times$.

# G   Ethical statement

SDEs are already a widely used modelling paradigm, primarily in fields such as finance and science. In this regard they are a tried-and-tested mathematical tool, with, to the best of the authors knowledge, no significant ethical concerns attached.

As this paper extends this existing methodology, then broadly speaking we expect the same to be true.

**Expected applications**   We anticipate the results of this paper as having applications to finance and the sciences. For example, to model the movement of asset prices, or to model predator-prey interactions.

As such no significant negative societal impacts are anticipated.

**Environmental impacts**   The primary contributions of this paper are speed improvements to existing methodologies. As such we anticipate a positive environmental impact from this paper, due to a reduction in the compute resources necessary to train model.

**Dataset content**   The data we are using contains no personally identifiable or offensive content.

**Dataset provenance**   All data used has been made publicly available, for example via the UCI machine learning repository [85]. To the best of our knowledge this availability was voluntary, and so we believe the use of the data to be ethical and without licensing issues.