# OpenReview forum: "Efficient and Accurate Gradients for Neural SDEs"
_NeurIPS.cc/2021/Conference — NeurIPS 2021 Poster_

### Official Review · Reviewer_Dmv1 · 2021-07-16

**Rating:** 6
**Confidence:** 3

**Summary:**

Neural SDE is an emerging tool for temporal modeling. This paper aims to improve the efficiency and accuracy of gradient calculation in the training of Neural SDEs. To this end, the authors introduce three techniques: (1) reversible Heun method that overcomes inconsistency in the forward and backward pass, (2) Brownian interval that speeds up sampling Brownian motion, and (3) weight clipping and specific activation to avoid gradient penalty in training. Through several numerical tests the authors demonstrate that these techniques altogether offer substantial improvements over the existing Neural SDE algorithms in speed and accuracy metrics.

**Limitations And Societal Impact:**

I think the clipping technique has some limitations that might be improved further. See the fourth comment above.

**Main Review:**

The paper aims to improves the efficiency and accuracy of training Neural SDEs. To this end, the authors start from the motivation and mathematical background, provide three techniques in order, and provide convincing numerical experiments to support the strengths. The whole paper is well organized and clearly written. The contribution to the open-source package will likely attract researchers and practitioners to adopt the proposed techniques and facilitate this process. Below are some concerns that should be clarified in a revision.

1. The authors should prove the reversibility of the Heurn method in Algorithm 1, although it is easy to verify.

2. I am confused by the result in Figure 2 where the midpoint and Heun methods "produce large errors decreasing with step size" (line 231). Can the authors provide some explanation? Does it conflict with the point in line 207 that "small or adaptive step sizes are necessary to obtain useful gradients (for standard numerical techniques)"?

3. The Brownian Interval is claimed to "offer memory efficiency, exact samples, and fast query times, all at once". To retrieve exact samples, do we need to keep the seeds at the root node the same for each pair of the forward pass and backward pass? In terms of memory efficiency, I feel the spatial complexity of the Brownian Interval is the same as storing Brownian motion because at each time step we need to store a random seed. The memory is saved because the size of the seed is much smaller than the Brownian motion increment. Is this understanding correct? By the way, in line 250, it should be "memory-efficient".

4. The clipping strategy seems the most ad-hoc technique in this paper. Clipping each entry to [-1/b, 1/b] is sufficient to ensure the Lipschitz constant no larger than 1, but it might be too restrictive such that approximation capacity is sacrificed. A more flexible alternative might be projecting according to the singular values. Please add more discussion on it. In addition, it is not so clear in Table 3 that clipping really helps. Could the authors provide the result of Reversible Heun with gradient penalty as well?

**Time Spent Reviewing:**

5

---

> ### Author Response · Authors · 2021-08-08
> **Response**
>
> We are glad to see the paper described as "well organised and clearly written".
>
> Moreover we are particularly happy to see the open-source contributions taken note of. We believe this is an important way to democratise techniques so as make them available to a broader community.
>
> **On the concerns that have been raised:**
>
> 1.
>
> We would be happy to include an explicit proof of the reversibility in a revision of the main paper. (It is already present in the Appendix.)
>
> 2.
>
> Apologies for the wording. To be clear, line 231 refers to the fact that
>
> (a) the errors decrease with step size;
>
> (b) at all step sizes, the errors are still very large compared to those of the reversible Heun method, introduced in this paper.
>
> This is consistent with the discussion on line 207, which states that decreasing step size is often necessary to obtain gradient errors sufficiently small for training to proceed.
>
> We will adjust this line to be sure that it is clear. We propose the following wording: "standard solvers produce errors decreasing with step size; however at all step sizes the error still remains relatively large". Does this seem reasonable to the reviewer?
>
> 3.
>
> The same seed is used for the forward and backward pass, yes. However we do _not_ store the random seed at each time step (and indeed doing so would not offer a meaningful improvement on the memory requirements).
>
> Instead, the random seeds at each time step are themselves regenerated, by splitting the random seed of the "parent" time step. (Which is itself recursively generated by splitting the random seed of _its_ parent time step, and so on.)
>
> This ensures that the memory requirements are at most the cost of storing (a) the top-level random seed and (b) the fixed-size LRU cache used to avoid repeated recursion. As both of these quantities are fixed then the memory requirements are truly $\mathcal{O}(1)$, rather than $\mathcal{O}(T)$.
>
> (Thank you for the correction on the typo.)
>
> 4.
>
> (a) No approximation capacity is sacrificed in the generator, as the clipping to $[-1/b,1/b]$ is applied only to the discriminator.
>
> (b) We did try projections according to singular values, and found this to be insufficient. Full singular value computations are relatively expensive; meanwhile approximate computations result in Lipschitz constants insufficiently close to 1 so as to avoid the exponential blow-up/decay.
>
> Some discussion of this is already included on line 328--329. We can add expand this discussion, so as to make this important point clear.
>
> (c) It is a technical limitation that we are unable to perform experiments on reversible Heun with gradient penalty. Reversible Heun is implemented as a custom backward pass, whilst gradient penalty requires a double-backward; that is to say it requires differentiating the backward pass itself.
>
> For consistency this double-backward must now be implemented as a double-continuous-adjoint. The interaction between this and the custom backward for reversible Heun is beyond the scope of what `torchsde` is currently equipped to handle.
>
> (d) On the topic of Table 3: we believe this unambiguously shows the benefit of clipping. Observe how midpoint with clipping obtains substantial improvements over midpoint with gradient penalty, across every test metrics that we consider.
>
> It is perhaps simply overshadowed by reversible Heun with clipping: this demonstrates further improvements, due to the further benefits provided by the reversible Heun method introduced in this paper.
>
> **Overall**
>
> We believe we have addressed every concern that the reviewer has raised, and where necessary we can adjust the paper accordingly.
>
> In light of this, and as the reviewer notes that our work offers "substantial improvements over the existing Neural SDE algorithms", we respectfully ask whether they would be willing to consider improving their review score?

---

### Official Review · Reviewer_Q4br · 2021-07-19

**Rating:** 6
**Confidence:** 2

**Summary:**

The submission studies neural SDEs, a more flexible version of classic SDEs with neural drifts and volatilities. Specifically, to improve the estimation/learning of such models, the submission provides two novel methods and one learning trick.

**Limitations And Societal Impact:**

The contributions of the submission on fitting neural SDE based on Stratonovich integrals are significant. However, I am curious whether explicit Euler method for Ito integrals also have some of the properties. In general, I have three questions.

1. Say we consider the SDE built on Ito integrals. Then, we use explicit Euler method as the SDE solver. With a loss function on the terminal state, can't we directly do backpropogation along the path? What is the difference between this naive method and the proposed one. Does (6) "exactly match" the gradient obtained by autodifferentiating the forward pass, in this naive approach?

2. Is building SDEs on Stratonovich integrals instead of Ito integrals a common practice for neural SDEs? In terms of model capacity, how do the two manners differ? Further, have you tried to apply the proposed methods to SDEs based on Ito-integrals as some kind of robust analysis?

3. If I understand it correctly, the method directly optimizes the drift and volatility coefficient functions. So, I am curious what the accuracy for estimating these functions might be e.g., the MSE for the drift estimation?

4. It seems that the results can also be applied to estimate structured parametric SDEs like the tasks in this paper (https://www.princeton.edu/~yacine/mle.pdf). (Not optimizing neural nets but functions with specific forms.) Is that correct? Would it be possible to provide comparisons with such closed-form methods?

**Main Review:**

The contributions of the submission are solid on improving the efficiency of fitting neural SDE based on Stratonovich integrals. The motivation is straightforward aiming at directly tackling the bottleneck of the gradient updates. Thorough empirical results are provided justifying the efficiency gain of the proposed methods.

**Time Spent Reviewing:**

2

---

> ### Author Response · Authors · 2021-08-08
> **Response**
>
> Thank you for your positive review. In answer to your questions:
>
> 1.
>
> Solving (6), whilst treating everything as an Ito integral, will not produce the correct gradients. No reversible solver is yet known to exist for Ito SDEs, and as a result there is no way to recover the same truncation errors.
>
> Indeed, the reversible Heun method -- a Stratonovich solver -- which is introduced in this paper, is to the best of our knowledge the first reversible SDE solver to have been invented.
>
> 2.
>
> The Neural SDE literature features both Ito and Stratonovich integrals. (For example [15] consider Ito whilst [16] consider Stratonovich.)
>
> Modelling capacity is known to be entirely unaffected. It is possible to transform Ito integrals to Stratonovich integrals, and vice-versa. For example [15] train Ito Latent SDEs whilst we train Stratonovich Latent SDEs, without any difference in performance.
>
> Within the machine learning literature this is possibly not yet a well-known fact, so we can certainly add a note explaining this.
>
> 3.
>
> One primary purpose of neural SDEs (over theoretically-constructed SDEs) is to model phenomena for which no theoretical description is known. When the "true" drift and volatility are unknown then we can obtain obtain no description of the error. So in some sense, the reviewer's question cannot be answered.
>
> The best that can be done is to train neural SDEs on synthetic data drawn from a known SDE, and in this regime the neural SDE is already known to recover the correct drift and diffusion; see either [16], or the SDE-GAN example of `torchsde`.
>
> 4.
>
> The reviewer is correct that traditional parametric SDEs may in principle be learnt using the techniques in this paper. We have deliberately avoided this comparison as one that we felt was of less interest to the NeurIPS community -- which is largely focused on deep learning, and problems that may only be tackled with deep learning. Space in the paper is limited, after all.
>
> We additionally note that in the simple scenario that a closed-form solution is known, then direct optimisation of the closed-form solution is generally preferable.
>
> **Overall**
>
> We hope we have addressed every concern the reviewer has. If all of the reviewer's concerns have been satisfactorily addressed, then we respectfully ask whether the reviewer would be willing to improve their score?

---

### Official Review · Reviewer_oQSS · 2021-08-03

**Rating:** 6
**Confidence:** 4

**Summary:**

The paper aims to address issues with training Neural SDEs; in particular, it aims to overcome issues relating to backpropagation through the SDE solve step. Previously this computation was marred by speed and accuracy issues arising due to high computational complexity, numerical errors in the SDEs solve step, as well as due to the high cost of reconstructing Brownian motion. To mitigate these issues, the paper introduces a variety of constructs. It introduces the reversible Heun method: an algebraically reversible SDE solver that reduces gradient errors to (effectively) zero. It also presents a Brownian interval construction to efficiently sample from and reconstruct Brownian motion. Additionally a weight-clipping procedure is described for improved training of neural SDEs as GANs. All of these contributions considerably advance the state-of-the-art in neural SDE training, as exemplified by various test tasks in the paper.

**Ethical Concerns:**

This paper is reasonably theoretical in nature; I have no ethical concerns about its impact.

**Limitations And Societal Impact:**

The authors have adequately addressed the limitations of their work as well as societal impact. There are separate sections in the discussion  about both, and Appendix G is entirely dedicated to this matter. It is noted that the work will likely have positive environmental impact, since it improves computational cost associated with training neural SDEs.

**Main Review:**

Strengths

1. The paper's contribution of a new SDE solver that is algebraically reversible (reversible Heun method) is novel. Moreover, the solver is shown to reduce numerical gradient error to near zero, which improves several test metrics on two datasets considerably over previous solvers (as exemplified by Table 1). A theorem is provided about strong convergence of the solver; the proof techniques are standard but well executed.

2. The Brownian interval construction is a well-suited data structure constructed for the task of efficiently sampling and reconstructing Brownian motion. From a data structures viewpoint, the construction is not particularly interesting (binary tree of (interval, seed) pairs), although it is carefully made to solve the problem of efficient sampling and reconstruction. Given the performance improvement showcased in Table 2 and Appendix F, I think it still makes for a good contribution to the neural SDE literature.

Weaknesses

1. Although the contribution regarding training of SDE-GANs via clipping as opposed to with gradient penalty did empirically help on the OU dataset as measured by a variety of test metrics (Table 3), I believe the motivation for clipping specifically is not particularly principled. I understand the need for a hard 1-Lipschitz constraint, but why not enforce the constraint via a Lie algebraic method like the one in "Cheap Orthogonal Constraints" [a] or a Bjorck orthonormalization approach like the one in "Sorting out Lipschitz function approximation" [b]? I believe both of these are more principled ways of maintaining a hard 1-Lipschitz constraint in contrast to clipping (which has had its own demonstrated set of problems since it was introduced in [50]). I would like to see a more principled discussion of the motivation for the clipping method in Section 5 and think that related work such as [a,b] should be addressed.

2. The baselines in the paper demonstrated the efficacy of the introduced approach to some degree, although I would have liked to see a more thorough comparison. For example, Table 1 provides exactly one baseline: the midpoint method. Why not at least provide the midpoint method together with (non-reversible) Heun? I don't believe this result is in the appendix either. I understand the field is reasonably niche, but this isn't an excuse for at the very least running Midpoint and Heun to compare with reversible Heun.

3. I think some of the discussion of results is lacking. In particular, I am thinking of Section 3.1. One the weights dataset, we see a training time improvement of roughly 2x, which is fully expected since reversible Heun requires only one function evaluation (per step) whereas midpoint and regular Heun require two. However, why does the benefit dwindle to 1.25x on the air quality dataset? I don't believe this is currently addressed.

Verdict

This work introduces the novel constructs of a reversible Heun SDE solver and a Brownian interval to considerably improve speed and accuracy of neural SDE training. The reversible Heun SDE solver is shown to work well relative to a single baseline on two datasets, although a fairly natural baseline (regular Heun) is omitted. The theory provided regarding convergence is well executed though makes use of largely standard proof techniques. The Brownian interval construction is effectively a well-applied binary-tree that greatly increases efficiency of sampling and reconstructing Brownian motion. The clipping approach to improving SDE-GAN training appears reasonable (and provides empirical benefit as exemplified by Table 3) but is not well-motivated with respect to existing methods of maintaining strict 1-Lipschitzness. Overall, the paper presents several practically useful contributions to the neural SDE literature and will benefit applicability, although I cannot currently accept it given my concerns above regarding various aspects of the paper.

Minor Corrections

The writing in the paper is, for the most part, good. Some minor corrections are given below:

L37: "By parameterising the drift and diffusion of an SDE as neural networks, then modelling capacity is greatly increased" -> ""By parameterising the drift and diffusion of an SDE as neural networks, modelling capacity is greatly increased""

L62: "computed via continuous adjoint method" -> "computed via the continuous adjoint method"

L67: "This is contrast" -> "This is in contrast" (perhaps your phrasing is common in British English; if so, please disregard)

L250: "memory-inefficient but time-intensive" -> "memory-efficient but time-intensive" (presumably this is what you meant to say)

References

[a] Casado, M.L., & Martínez-Rubio, D. (2019). Cheap Orthogonal Constraints in Neural Networks: A Simple Parametrization of the Orthogonal and Unitary Group. ArXiv, abs/1901.08428.

[b] Anil, C., Lucas, J., & Grosse, R.B. (2019). Sorting out Lipschitz function approximation. ICML.

### Post-rebuttal Update

I have improved my score from a 5 to a 6.

**Time Spent Reviewing:**

6 hours

---

> ### Author Response · Authors · 2021-08-08
> **Response**
>
> **On the strengths:**
>
> Thank you for your review. We are glad to hear that our contributions _"considerably improve speed and accuracy of neural SDE training"_, and that _"the theory provided regarding convergence is well executed"_.
>
> We are particularly glad to read the emphatically positive statement _"these contributions considerably advance the state-of-the-art in neural SDE training"_.
>
> **On the weaknesses:**
>
> 1.
>
> We would begin by respectfully disagreeing that the current approach is not "principled". It is simple; that does not make it unprincipled.
>
> Moreover, whilst other approaches to maintaining the Lipschitz constraint may certainly be considered, we would note that both Lie exponential maps and Björck orthonormalisation are truncated approximations. This need not produce a hard 1-Lipschitz constraint, and at minimum requires additional tricks to make work.
>
> For example, a first attempt might be to introduce a carefully tuned coefficient $\alpha \in (0, 1)$, to scale each linear transformation. In practice we have already tried this (at some length with spectral normalisation), and observed that due to the exponential scaling over the duration of the integration, it was impossible to select a value of $\alpha$ that did not either result in a Lipschitz constant either decaying to zero or exploding to infinity.
>
> We would nonetheless be very happy to include [a, b] as additional references, and as much of the above discussion as space allows, so as to more strongly motivate the choices presented in the paper.
>
> 2.
>
> We agree this is a reasonable point. We actually did run experiments with Heun, and found that the results were essentially identical to midpoint. As such they were left out to avoid duplication.
>
> We agree that it would make sense to add them nonetheless, and would be happy to do so in a camera-ready submission.
>
> 3.
>
> The benefit is smaller because of differing overheads. The time series of the weights dataset are twice as long as the time series of the air quality dataset, so that the integration of the SDE is a greater proportion of the overall computational cost. Fixed overheads in both regimes include the generation of the initial value, the updating of parameters, and so on.
>
> We will include a note explaining this.
>
> **Overall**
>
> We believe we have successfully addressed every concern that the reviewer has raised.
>
> Moreover we note that the reviewer has made several substantially positive comments about this paper -- we particularly highlight how the multiple contributions of this paper all "considerably advance the state-of-the-art". As such, we respectfully ask whether the reviewer would be willing to improve their score?

---

> > ### Comment · Reviewer_oQSS · 2021-08-29
> > **Response**
> >
> > I appreciate the response to my concerns; I believe they are for the most part, addressed. Please include these clarifications in the final version of the paper. I've also reviewed the other reviews and the corresponding responses. Overall, I am leaning towards acceptance and see the paper being a positive contribution to the conference (contingent on the addition of the above clarifications); hence, I have decided to update my score from a 5 to a 6.

---

### Official Review · Reviewer_e5Kh · 2021-08-08

**Rating:** 7
**Confidence:** 4

**Summary:**

This paper studies both neural SDE and construction of Brownian intervals. The neural SDE considered in the current paper is in a more general form comparing to other recent works where diffusion coefficients are constant. The Brownian interval seems to provide efficient sampling algorithm for Brownian motion which could be used in the discretized algorithm.

**Main Review:**

Positive:

1. The Neural SDE considered in this paper is very general. The Stratonovich SDE and the adjoint equation are nicely used in this framework. The reversible Heun method and the algorithm seems to be well explained within the Stratonovich SDE.

2. A direct discretization of Stratonovich SDE would require to solve an algebraic equation in order to repeat the iteration. The Algorithm for the forward and backward process introduce an intermediate process to avoid this step. This idea seems to be very interesting.

3. The Brownian interval idea following [15] is very interesting. The memory cost of this algorithm seems to be promising.

4. The experiment shows impressive and supportive results of the algorithm.

Minor point:

1. It would be good to provide the intuition and the reason why do you define Heun method in Definition 1. What is the relation between this definition and the discretization of Stratonovich SDE.




**Time Spent Reviewing:**

8

---

> ### Author Response · Authors · 2021-08-08
> **Response**
>
> We thank the reviewer for their positive review, and the multiple positive remarks they make about this paper.
>
> **Minor point**
>
> With respect to the one minor point raised: assuming the reviewer is referring to Definition D.1, we agree that this can be done.
>
> We propose to
>
> 1. add the following wording: "The values $Y_n$ numerically approximate the solution $y_{nh}$.";
> 2. move equations (19)--(20) up a little further, as these are what provide the intuition of the solution $Y_n$ and the auxiliary variable $Z_n$.
>
> **Overall**
>
> If this is satisfactory to the reviewer we hope they will be willing to maintain or improve their positive review score.

---

### Decision · Program_Chairs · 2021-09-27

**Decision:**

Accept (Poster)

**Comment:**

After reading the author responses and discussing with the authors, the reviewers have reached a consensus that this paper should be accepted. I concur with that judgement.  The accuracy improvements proposed by this method seem significant and useful across a range of tasks. And the mathematical insights on which the result are based will be interesting to even those in the community who aren't using neural SDEs.